# DGS-Net: Distillation-Guided Gradient Surgery for CLIP Fine-Tuning in AI-Generated Image Detection

**Jiazhen Yan** [1]  **Ziqiang Li** [1]  **Fan Wang** [2]  **Boyu Wang** [1]  **Ziwen He** [1]  **Zhangjie Fu** [1]

## Abstract

The rapid progress of generative models such as GANs and diffusion models has led to the widespread proliferation of AI-generated images, raising concerns about misinformation and trust erosion in digital media. Although large-scale multimodal models like CLIP offer strong transferable representations for detecting synthetic content, fine-tuning them often induces catastrophic forgetting, which degrades pre-trained priors and limits cross-domain generalization. To address this issue, we propose the Distillation-guided Gradient Surgery Network (DGS-Net), a novel framework that preserves transferable pre-trained priors while suppressing task-irrelevant components. Specifically, we introduce a gradient-space decomposition that separates harmful and beneficial descent directions during optimization. By projecting task gradients onto the orthogonal complement of harmful directions and aligning with beneficial ones distilled from a frozen CLIP encoder, DGS-Net achieves unified optimization of prior preservation and irrelevant suppression. Extensive experiments on 50 generative models demonstrate that our method outperforms state-of-the-art approaches by an average margin of 6.6%, achieving superior detection performance and generalization. Our code is available at https://horizontel.github.io/DGS-Net/.

## 1. Introduction

The rapid advancement of deep learning has driven significant progress in generative models such as generative adversarial networks (GANs) (Karras et al., 2018; Park et al., 2019) and diffusion models (Rombach et al., 2022; Wu et al.,

---

[1]School of Computer Science, Nanjing University of Information Science and Technology [2]University of Macau. Correspondence to: Zhangjie Fu <fzj@nuist.edu.cn>.

*Proceedings of the 43rd International Conference on Machine Learning*, Seoul, South Korea. PMLR 306, 2026. Copyright 2026 by the author(s).

2024). Owing to their low generation cost and high visual fidelity, they are widely applied across various domains. However, their widespread use has also raised serious concerns about privacy violations, misinformation dissemination, and the erosion of trust in digital media (Wang et al., 2024; 2025; Lin et al., 2025c; Hu et al., 2026; Wang et al., 2026a;b). As a result, developing reliable detectors to verify the authenticity of AI-generated images has become an urgent task. Despite notable progress in recent detection methods (Li et al., 2025a; Lin et al., 2025a;b; Liu et al., 2025; 2026b), achieving robust generalization to unseen generative techniques remains unsolved.

Large-scale multimodal pre-trained models, such as CLIP (Radford et al., 2021), have recently emerged as powerful tools for detecting AI-generated images. By leveraging transferable representations learned from massive image–text datasets, CLIP constructs a semantic embedding space with strong open-set scalability and cross-domain generalization (Tan et al., 2025; Yan et al., 2024b), providing a robust foundation for identifying synthetic images (Ojha et al., 2023) produced by diverse generative models. However, CLIP's pre-training objective is not explicitly optimized to capture generation artifacts. To address this limitation, recent studies have adapted CLIP for detection tasks through parameter-efficient tuning strategies (Liu et al., 2024a; Tan et al., 2025; Fu et al., 2025) such as LoRA (Hu et al., 2022). Despite these efforts, fine-tuning often induces **catastrophic forgetting**, eroding the transferable priors and geometric structure of the pre-trained embedding space. Consequently, the adapted model may overfit to dataset-specific patterns rather than learning artifact-level features that generalize across different generative mechanisms.

To investigate this phenomenon, we construct four datasets using ProGAN (Karras et al., 2018), R3GAN (Huang et al., 2024b), SDXL (Podell et al., 2023), and SimSwap (Chen et al., 2020), each containing both real and generated images across multiple categories. We then extract features using three variants of CLIP: the original model (Ojha et al., 2023), a LoRA fine-tuned model, and our proposed method. The extracted features are visualized using t-SNE, as shown in Figure 1. The frozen CLIP model preserves the geometric structure and local coherence learned during pre-training but

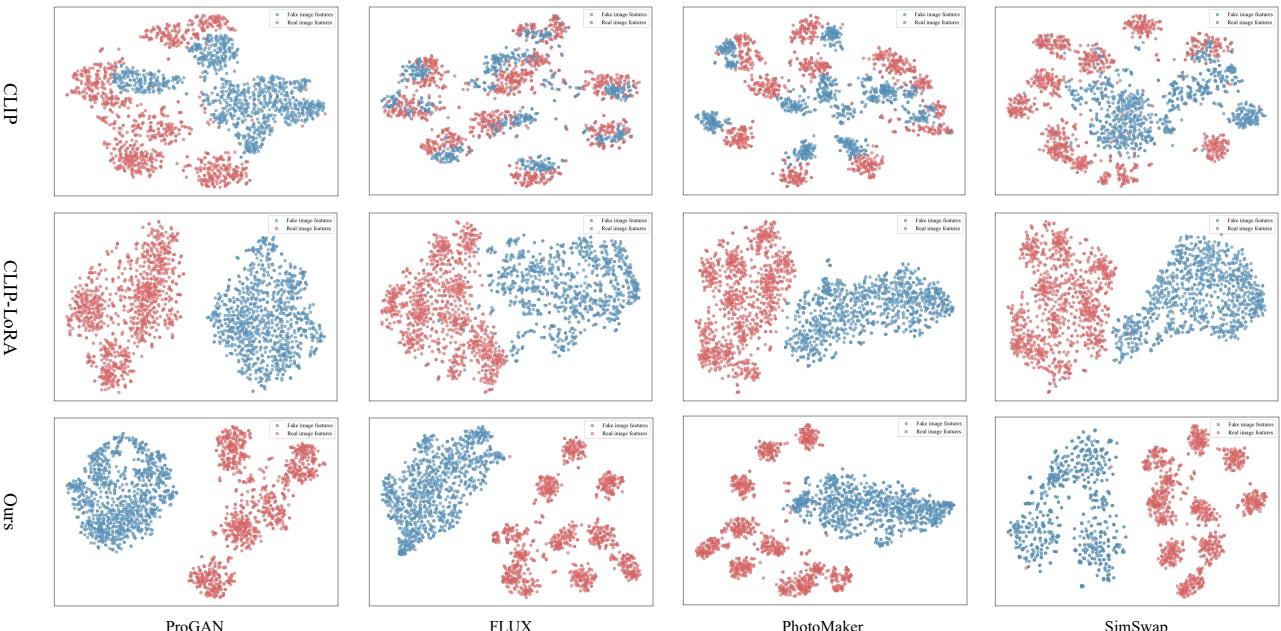

*Figure 1.* **T-SNE Visualization of Features Extracted Using CLIP, CLIP-LoRA and Ours**. Our method achieves strong real/fake discrimination while simultaneously preserving the prior knowledge embedded in the pre-trained model.

exhibits weak real/fake separability. In contrast, LoRA fine-tuning enhances separability but collapses the representation geometry and degrades cross-domain detection performance. These observations indicate that catastrophic forgetting in existing fine-tuning strategies compromises the transferable priors of the pre-trained model.

To address these issues, we introduce a knowledge distillation framework (Hinton et al., 2015) tailored for CLIP-based detectors. As illustrated in Figure 1, the pre-trained CLIP features inherently contain numerous components unrelated to generation artifacts. Conventional feature distillation methods, which enforce global alignment between teacher and student representations, may inadvertently retain these task-irrelevant factors. Therefore, our goal is to **preserve transferable pre-trained priors** while **suppressing task-irrelevant components**. Specifically, we introduce a gradient-space decomposition strategy, where the positive half-space of task gradients is defined as harmful directions and the negative half-space as beneficial directions (see Section 3 for details). Building on this principle, we propose a **D**istillation-guided **G**radient **S**urgery **Net**work (**DGS-Net**). During backpropagation, we first project the image gradients of the training network onto the orthogonal complement of the harmful directions estimated from text gradients, effectively suppressing components irrelevant to the detection task. Meanwhile, we exploit the beneficial descent directions extracted from a frozen CLIP image encoder to guide lightweight alignment of the training network's representations, thereby preserving the transferable priors acquired during pre-training. As shown in Figure 1, our approach

achieves strong real/fake discrimination while maintaining the pre-trained model's prior knowledge, ultimately enhancing generalization across diverse generative models.

In summary, our work makes following key contributions:

- To the best of our knowledge, this is the first to systematically diagnose catastrophic forgetting induced by CLIP fine-tuning in AI-generated image detection. We further introduce a novel gradient-space decomposition that disentangles CLIP representations into transferable pre-trained priors and task-irrelevant components.

- We propose an innovative detection framework, **DGS-Net**, which employs distillation-guided gradient decoupling and alignment to preserve transferable pre-trained priors while suppressing task-irrelevant knowledge, thereby improving both detection accuracy and cross-domain generalization.

- Extensive experiments across 50 diverse generative models demonstrate that our method consistently outperforms state-of-the-art approaches, achieving an average improvement of 6.6% in detection accuracy, highlighting its robustness and universality.

## 2. Related Works

### 2.1. AI-Generated Image Detection

With the rapid advancement of generative models such as GANs (Karras et al., 2018; 2021; Huang et al., 2024b) and

diffusion models (Ho et al., 2020; Dhariwal & Nichol, 2021; Wu et al., 2024), making it increasingly difficult for the human eye to distinguish real images from generated ones. Extensive research has been devoted to AI-generated image detection, which can be broadly categorized into **artifact-based methods** (Zheng et al., 2024; Tao et al., 2025; Zou et al., 2025a;b; Zhong et al., 2026) and **data-centric methods** (Wang et al., 2020; Chen et al., 2025). Artifact-based approaches typically exploit low-level forensic cues that expose generation traces, such as frequency irregularities (Luo et al., 2021; Tan et al., 2024a; Yan et al., 2026), pixel-level inconsistencies (Chen & Yang, 2021; Cavia et al., 2024), gradient patterns (Tan et al., 2023), neighboring-pixel dependencies (Tan et al., 2024b), and reconstruction errors (Wang et al., 2023; Chu et al., 2025; Chen et al., 2025). For example, BiHPF (Jeong et al., 2022) enhances artifact visibility by applying dual high-pass filters; NPR (Tan et al., 2024b) reinterprets upsampling operations to construct an efficient artifact representation; while DIRE (Wang et al., 2023) captures generation discrepancies via reconstruction error analysis. In contrast, data-centric methods aim to learn more generalizable artifact representations through large-scale data and augmentation strategies. Representative efforts include data augmentation (Wang et al., 2020; Li et al., 2025b) and dataset construction (Chen et al., 2024a; Guillaro et al., 2025; Chen et al., 2025; Zhou et al., 2025a). SAFE (Li et al., 2025b) integrates cropping and augmentations such as ColorJitter, RandomRotation, and Random-Mask to improve generalization, while B-Free (Guillaro et al., 2025) mitigates dataset bias using self-conditioned inpainted reconstructions and content-aware augmentation.

With the rapid rise of large pre-trained models, particularly vision–language models such as CLIP (Radford et al., 2021), recent studies have begun exploiting their powerful feature extraction and semantic understanding capabilities for AI-generated image detection. The impressive generalization ability of CLIP demonstrated in UnivFD (Ojha et al., 2023) has inspired a series of follow-up works, which can be broadly divided into two categories: (1) **feature-based methods**, which freeze the pre-trained network and optimize only the extracted features to enhance generalization (Ojha et al., 2023; Koutlis & Papadopoulos, 2024; Zhang et al., 2025; Zhou et al., 2025b); and (2) **fine-tuning-based methods**, which update model parameters to learn task-specific features through full fine-tuning, low-rank adaptation (LoRA) (Liu et al., 2024b; Tan et al., 2025; Yan et al., 2025a;b; Liu et al., 2026a), or lightweight adaptation modules (Liu et al., 2024a; Yan et al., 2024b; Tao et al., 2025). For instance, VIB-Net (Zhang et al., 2025) applies a Variational Information Bottleneck to decouple CLIP-extracted features, effectively suppressing task-irrelevant information. C2P-CLIP (Tan et al., 2025) injects category-consistent cues into the text encoder to improve cross-modal alignment. Ef-

fort (Yan et al., 2024b) constructs two orthogonal subspaces to preserve pre-trained priors while learning forgery-related representations. NS-Net (Yan et al., 2025b) constructs a NULL-Space of semantic features to remove semantic interference embedded in visual features.

Despite these advances, existing fine-tuning strategies often degrade the transferable priors of pre-trained models, leading the network to rely on dataset-specific shortcuts rather than learning intrinsic artifact-level cues that generalize across different generators.

## 2.2. Knowledge Distillation

Knowledge Distillation (Hinton et al., 2015) is a powerful technique for transferring knowledge from a large teacher model to a smaller student model. Numerous studies (Li et al., 2017; Wang et al., 2019; Jia et al., 2024) focus on selecting the most valuable features for knowledge distillation in downstream tasks, with the goal of preserving the richest prior knowledge. This technique has been successfully applied across a variety of domains, including visual recognition (Kim et al., 2018; Li et al., 2021; 2023; Huang et al., 2024a), language model compression (Jiao et al., 2019; Park et al., 2021; Wu et al., 2023), and multimodal representation learning (Fang et al., 2021; Chen et al., 2023; 2024b). A seminal work (Chen et al., 2017) introduced the first distillation framework for object detection, combining feature imitation with prediction mimicking. (Pfeiffer et al., 2020) incorporated a distillation loss into adapter-based fine-tuning to transfer representations from pre-trained models, preventing feature drift during adaptation. (Wei et al., 2023) proposed feature-level knowledge distillation during fine-tuning, which helps the student model maintain cross-modal semantic alignment while learning downstream tasks.

However, our research shows that not all pre-trained knowledge is beneficial for downstream adaptation, as some representations can negatively impact task performance. In response, we extend the traditional distillation paradigm to selectively retain useful priors while suppressing irrelevant, harmful representations. Unlike conventional feature distillation methods (Tian et al., 2019; Miles et al., 2021; Chen et al., 2021; Yang et al., 2021), which penalize discrepancies at the representation level, our approach operates in the gradient space. It suppresses gradients that are irrelevant to the task while injecting a small, beneficial knowledge prior into the descent direction, ultimately enhancing the performance of AI-generated image detection.

## 3. Preliminaries

Before introducing our method, we first establish a directional convention for interpreting the gradient dynamics of classification losses. For any differentiable classification

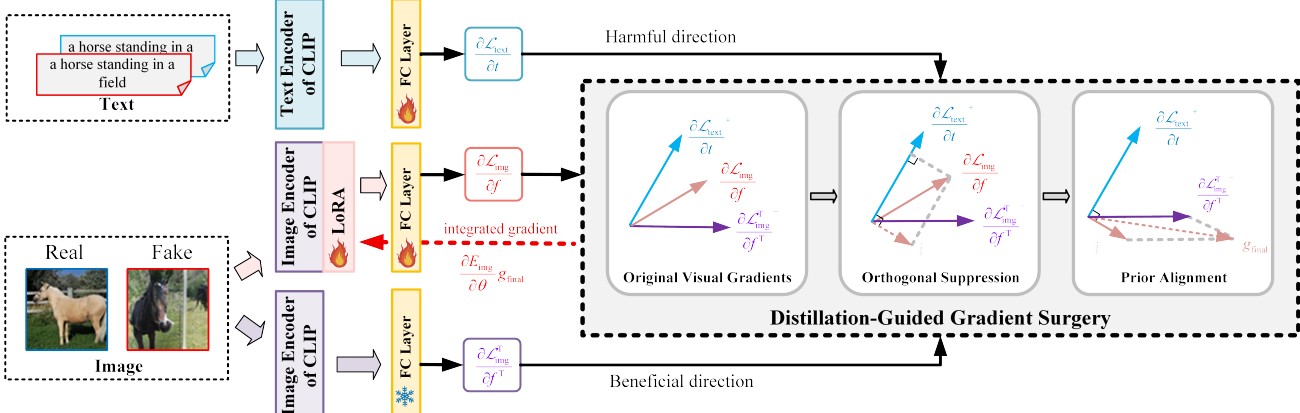

*Figure 2.* **Overview of the proposed Distillation-guided Gradient Surgery Network (DGS-Net).** We introduce a gradient-space decomposition that separates harmful and beneficial descent directions during optimization. What's more, it consists of two core components: Orthogonal Suppression and Prior Alignment, which aim to suppress task-irrelevant representations and preserve transferable priors established during large-scale pre-training, substantially enhancing the generalization performance of AI-generated image detection.

loss, the first-order gradient of the representation indicates the direction of the steepest local variation in the loss landscape (Kingma, 2014). Specifically, a positive gradient value at a given coordinate implies that a positive perturbation along that direction increases the loss (i.e., a **harmful direction**), whereas a negative gradient value implies that a positive perturbation decreases the loss (i.e., a **beneficial direction**).

Formally, let the representation vector be denoted as $u \in \mathbb{R}^d$, and consider a binary classification loss function (*e.g.*, BCE-WithLogits) represented by $\mathcal{L}(u, y)$, where $y \in \{0, 1\}$. For any smooth classification loss, the gradient $\nabla_u \mathcal{L}$ indicates the steepest ascent direction of $\mathcal{L}$ in the representation space. The corresponding first-order directional derivative can be approximated as

$$\mathcal{L}(u + \varepsilon e, y) \approx \mathcal{L}(u, y) + \varepsilon \langle \nabla_u \mathcal{L}(u, y), e \rangle, \quad (\varepsilon \to 0^+), \tag{1}$$

where $e$ denotes a unit direction vector. This formulation naturally motivates the following definitions.

**Harmful direction.** A direction is considered harmful if an infinitesimal positive perturbation along it increases the loss. For a canonical basis vector $e_j$,

$$\frac{\partial \mathcal{L}}{\partial u_j} > 0 \Leftrightarrow \mathcal{L}(u + \varepsilon e_j, y) > \mathcal{L}(u, y) \quad (\varepsilon \to 0^+), \tag{2}$$

where $u_j = \langle u, e_j \rangle$. Thus, coordinates with $\frac{\partial \mathcal{L}}{\partial u_j} > 0$ correspond to harmful components at the current point. In vector form, we define the positive part of the gradient as

$$g^+ \triangleq \left[ \nabla_u \mathcal{L} \right]_+, \quad ([a]_+)_j = \max(a_j, 0), \tag{3}$$

so that $g^+$ aggregates all harmful coordinates into a unified composite direction.

**Beneficial direction.** Analogously, a beneficial direction corresponds to a perturbation that locally decreases the loss. We define the negative part of the gradient as

$$g^- \triangleq \left[ \nabla_u \mathcal{L} \right]_-, \quad ([a]_-)_j = \min(a_j, 0), \tag{4}$$

so that $g^-$ captures the composite direction that induces a local loss reduction under a positive perturbation.

**Summary.** In summary, $g^+$ characterizes the local half-space of coordinates whose activations should be suppressed, whereas $g^-$ identifies the complementary half-space that can be encouraged to facilitate optimization.

## 4. Methodology

The overall framework of the proposed DGS-Net is depicted in Figure 2. It consists of two core components: **Orthogonal Suppression** and **Prior Alignment**. These modules respectively aim to suppress task-irrelevant representations and preserve transferable priors established during large-scale pre-training. Specifically, in *Orthogonal Suppression*, the image gradients of the training network are orthogonally projected onto the subspace complementary to the harmful directions inferred from text gradients, thereby mitigating cross-modal interference. In *Prior Alignment*, the coordinate-wise negative components of the frozen CLIP image gradients are introduced as lightweight alignment signals to reinforce beneficial pre-trained priors. The subsequent section provides a detailed methodological description, elaborating on the design principles and key mechanisms that underlie DGS-Net.

### 4.1. Notation and Setup

Given image–label pair $(x, y)$ with $y \in \{0, 1\}$, where $y = 1$ denotes a *fake* image and $y = 0$ denotes a *real* image, CLIP

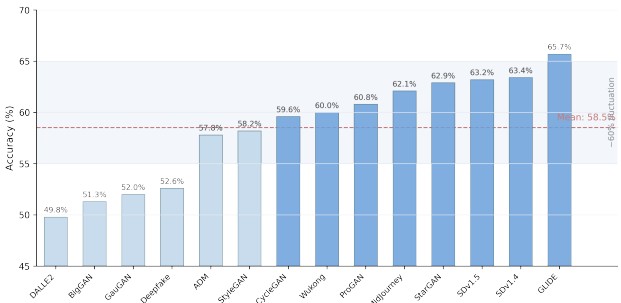

*Figure 3.* **Classification using only text descriptions achieves an accuracy of approximately 60%.** We used BLIP to convert images into textual descriptions and trained the detector directly on these text inputs. The resulting detection accuracy fluctuated around 60%, indicating that semantic information is partially correlated with the labels. However, most of them act as distractors that hinder cross-generator generalization.

provides an image encoder $E_{\text{img}}$ and a text encoder $E_{\text{text}}$. The corresponding feature extraction processes are defined:

$$f \;=\; E_{\text{img}}(x;\theta) \in \mathbb{R}^d, \quad t \;=\; E_{\text{text}}(z;\phi) \in \mathbb{R}^d, \quad (5)$$

where $\theta$ denotes the trainable parameters of the image encoder, and $\phi$ is frozen since the text encoder is only used to extract semantic representations. Here, $z$ represents the text description associated with image $x$. In this work, we incorporate LoRA (Low-Rank Adaptation) layers as the trainable parameters $\theta$ for fine-tuning CLIP's image encoder, while using BLIP (Li et al., 2022) to automatically generate the corresponding text description $z$. Additionally, we maintain a frozen image encoder $E_{\text{img}}^T$—a copy of the CLIP image encoder prior to fine-tuning—to extract reference features:

$$f^T \;=\; E_{\text{img}}^T(x) \in \mathbb{R}^d. \quad (6)$$

We employ the BCEWithLogits loss for all branches, formulated as:

$$\mathcal{L}_{\text{img}} = \ell\big(h_{\text{img}}(f), y\big), \quad \mathcal{L}_{\text{text}} = \ell\big(h_{\text{text}}(t), y\big), \quad (7)$$
$$\mathcal{L}_{\text{img}}^T = \ell\big(h_{\text{img}}^T(f^T), y\big),$$

where $h_{\text{img}}$, $h_{\text{text}}$, and $h_{\text{img}}^T$ denote linear classification heads. The corresponding gradients with respect to the feature representations are given by

$$g_{\text{task}} \;=\; \nabla_f \mathcal{L}_{\text{img}} \in \mathbb{R}^d, \quad g_{\text{text}} \;=\; \nabla_t \mathcal{L}_{\text{text}} \in \mathbb{R}^d, \quad (8)$$
$$g_{\text{img}} \;=\; \nabla_{f^T} \mathcal{L}_{\text{img}}^T \in \mathbb{R}^d.$$

### 4.2. Orthogonal Suppression

To effectively suppress task-irrelevant components, it is crucial to first characterize these irrelevant features. Large-scale pre-trained models exhibit strong semantic representation

capabilities, as evidenced by their performance on ImageNet classification. Prior studies have explored different ways to handle such semantics: some works (Yan et al., 2024b; Tan et al., 2025) exploit them to enhance model generalization, whereas others (Yan et al., 2025b; Zhang et al., 2025) seek to eliminate them in order to emphasize low-level artifact cues. To further investigate the influence of semantic features on AI-generated image detection, we convert each image into a textual description and perform binary classification using only the text features extracted by the text encoder. As illustrated in Figure 3, this approach achieves an accuracy of approximately 60%, suggesting that semantic information is partially correlated with the labels. However, the majority of these semantics act as distractors, which may hinder generalization. Therefore, we explicitly characterize the semantic priors embedded in CLIP's visual encoder and only suppress the harmful semantic components from pre-training, which interfere with the reliable detection of AI-generated images.

Specifically, we extract the harmful gradients from the text branch to suppress pre-trained components that are irrelevant to the detection task. As discussed in Section 3, if the text-branch gradient $g_{\text{text}}$ is positive in a given dimension, a small positive perturbation along that coordinate will increase the loss. Consequently, gradient descent will update the corresponding semantic coordinate in the negative direction to reduce the loss. However, some update directions, which are often associated with detection errors, may act as harmful directions that conflict with genuine forensic cues, ultimately impairing cross-generator generalization. To address this issue, we regard the positive half-space of the text gradient as a **harmful direction** that should be suppressed, formulated as:

$$g_{\text{harm}} \triangleq g_{\text{text}}^+ = \big[\nabla_t \mathcal{L}_{\text{text}}\big]_+ \in \mathbb{R}^d. \quad (9)$$

We then seek an update direction $\tilde{g}$ for the image representation that excludes the component aligned with $g_{\text{harm}}$. Following the principle of minimum deviation, this can be expressed as an optimization problem with an orthogonality constraint:

$$\Delta g^\star = \arg\min_{\Delta g} \; \tfrac{1}{2}\|\Delta g - g_{\text{task}}\|_2^2 \quad \text{s.t.} \quad \langle \Delta g, \hat{g}_{\text{harm}} \rangle = 0,$$
$$\hat{g}_{\text{harm}} = \frac{g_{\text{harm}}}{\|g_{\text{harm}}\|_2}. \quad (10)$$

The closed-form Lagrangian solution is given by

$$\tilde{g} \triangleq \Delta g^* = \big(I - \hat{g}_{\text{harm}}\hat{g}_{\text{harm}}^\top\big)\, g_{\text{task}}, \quad (11)$$

where the projection operator $(I - \hat{g}_{\text{harm}}\hat{g}_{\text{harm}}^\top)$ removes the harmful component, ensuring that the update direction is orthogonal to the undesired semantic subspace.

After applying orthogonal suppression, the gradient update direction shifts from $g_{\text{task}}$ to $\tilde{g}$, effectively removing harmful

*Table 1.* **Cross-model Accuracy (Acc.) Performance on the AIGCDetectBench (Zhong et al., 2023) Dataset.** The first column represents the accuracy of detecting real images (R.Acc.), and the others are the accuracy of detecting fake images (F.Acc.).

| Detection Method | Real Image | Generative Adversarial Networks | | | | | | | Other | | Diffusion Models | | | | | | | | mAcc. |
|---|---|---|---|---|---|---|---|---|---|---|---|---|---|---|---|---|---|---|---|
| | | Pro-GAN | Cycle-GAN | Big-GAN | Style-GAN | Style-GAN2 | Gau-GAN | Star-GAN | WFIR | Deep-fake | SDv1.4 | SDv1.5 | ADM | GLIDE | Mid-journey | Wukong | VQDM | DALLE2 | |
| CNN-Spot (Wang et al., 2020) | 99.0 | 95.3 | 18.7 | 1.8 | 36.5 | 22.0 | 2.5 | 23.1 | 1.1 | 29.3 | 55.9 | 55.6 | 1.8 | 4.8 | 5.2 | 27.6 | 0.7 | 4.5 | 29.0 |
| UnivFD (Ojha et al., 2023) | 92.3 | 98.9 | 90.5 | 79.2 | 55.7 | 48.7 | 91.1 | 96.9 | 92.8 | 26.9 | 96.3 | 96.0 | 12.7 | 75.6 | 61.2 | 84.7 | 45.6 | 62.3 | 72.7 |
| FreqNet (Tan et al., 2024a) | 89.9 | 99.4 | 57.1 | 51.0 | 75.1 | 67.5 | 9.9 | 88.4 | 95.4 | 35.8 | 99.9 | 99.8 | 37.7 | 78.9 | 80.8 | 98.0 | 34.1 | 88.8 | 71.7 |
| NPR (Tan et al., 2024b) | 99.3 | 98.9 | 29.3 | 16.5 | 67.1 | 58.7 | 1.7 | 6.5 | 7.9 | 0.1 | 100.0 | 99.9 | 26.5 | 69.2 | 71.0 | 97.7 | 15.4 | 89.8 | 53.1 |
| Ladeda (Cavia et al., 2024) | 99.8 | 99.7 | 7.2 | 29.0 | 95.6 | 98.3 | 4.9 | 0.0 | 19.2 | 0.1 | 100.0 | 99.9 | 27.3 | 79.7 | 88.8 | 97.9 | 15.5 | 92.4 | 58.6 |
| AIDE (Yan et al., 2024a) | 93.0 | 95.3 | 88.0 | 96.9 | 89.6 | 97.0 | 90.0 | 97.0 | 42.9 | 9.2 | 99.8 | 99.7 | 92.4 | 98.7 | 63.7 | 98.9 | 90.1 | 98.9 | 85.6 |
| C2P-CLIP* (Tan et al., 2025) | 99.8 | 100.0 | 99.8 | 99.8 | 72.2 | 78.0 | 87.7 | 100.0 | 34.9 | 67.6 | 100.0 | 99.9 | 50.5 | 73.3 | 28.4 | 99.7 | 93.3 | 98.1 | 82.4 |
| DFFreq (Yan et al., 2026) | 98.0 | 98.0 | 70.9 | 93.6 | 99.0 | 98.9 | 93.8 | 97.4 | 3.5 | 76.0 | 99.9 | 99.8 | 65.9 | 87.8 | 94.8 | 99.1 | 76.0 | 95.9 | 85.2 |
| SAFE (Li et al., 2025b) | 99.2 | 99.7 | 85.6 | 83.5 | 88.1 | 96.7 | 89.0 | 99.9 | 8.4 | 17.3 | 99.8 | 99.6 | 36.8 | 90.5 | 86.3 | 98.5 | 84.0 | 92.0 | 80.8 |
| Effort (Yan et al., 2024b) | 99.8 | 100.0 | 100.0 | 100.0 | 79.6 | 82.1 | 100.0 | 100.0 | 99.9 | 76.6 | 100.0 | 99.9 | 53.1 | 81.2 | 44.2 | 99.9 | 96.3 | 75.8 | 88.1 |
| NS-Net (Yan et al., 2025b) | 97.7 | 100.0 | 99.9 | 100.0 | 95.7 | 98.6 | 98.5 | 100.0 | 68.9 | 60.6 | 100.0 | 99.9 | 85.3 | 97.2 | 77.4 | 100.0 | 98.9 | 98.8 | 93.2 |
| Ours | 95.3 | 100.0 | 100.0 | 100.0 | 99.5 | 99.8 | 100.0 | 100.0 | 84.6 | 96.7 | 100.0 | 99.9 | 95.2 | 99.0 | 88.0 | 100.0 | 99.7 | 99.6 | 97.6 |

*Table 2.* **Cross-model Average Precision (A.P.) Performance on the AIGCDetectBench (Zhong et al., 2023) Dataset.**

| Detection Method | Generative Adversarial Networks | | | | | | | Other | | Diffusion Models | | | | | | | | mA.P. |
|---|---|---|---|---|---|---|---|---|---|---|---|---|---|---|---|---|---|---|
| | Pro-GAN | Cycle-GAN | Big-GAN | Style-GAN | Style-GAN2 | Gau-GAN | Star-GAN | WFIR | Deep-fake | SDv1.4 | SDv1.5 | ADM | GLIDE | Mid-journey | Wukong | VQDM | DALLE2 | |
| CNN-Spot(Wang et al., 2020) | 99.9 | 91.0 | 59.2 | 94.8 | 94.2 | 86.1 | 82.5 | 65.5 | 74.2 | 97.6 | 97.6 | 66.6 | 83.1 | 80.4 | 88.9 | 57.0 | 87.8 | 84.3 |
| UnivFD (Ojha et al., 2023) | 99.9 | 96.2 | 94.7 | 90.9 | 91.1 | 97.6 | 99.7 | 83.0 | 83.5 | 99.3 | 99.1 | 65.2 | 95.3 | 91.7 | 97.4 | 87.2 | 89.8 | 91.9 |
| FreqNet (Tan et al., 2024a) | 100.0 | 97.5 | 87.9 | 95.9 | 94.2 | 54.0 | 100.0 | 57.0 | 88.5 | 99.9 | 99.9 | 73.9 | 94.1 | 94.4 | 99.1 | 74.9 | 92.3 | 88.4 |
| NPR (Tan et al., 2024b) | 100.0 | 98.6 | 79.4 | 98.6 | 99.6 | 71.3 | 97.6 | 65.5 | 92.5 | 100.0 | 99.9 | 74.1 | 97.4 | 97.8 | 99.9 | 81.5 | 99.3 | 91.4 |
| Ladeda (Cavia et al., 2024) | 100.0 | 99.0 | 89.5 | 100.0 | 100.0 | 96.4 | 90.3 | 93.6 | 92.6 | 95.5 | 95.8 | 84.8 | 96.6 | 93.5 | 93.0 | 84.5 | 93.9 | 94.0 |
| AIDE (Yan et al., 2024a) | 99.6 | 98.2 | 91.6 | 99.4 | 99.9 | 74.1 | 99.7 | 90.8 | 66.5 | 99.9 | 99.9 | 99.4 | 99.9 | 90.3 | 99.9 | 98.9 | 99.9 | 94.5 |
| C2P-CLIP* (Tan et al., 2025) | 100.0 | 100.0 | 99.9 | 99.3 | 99.7 | 98.8 | 100.0 | 99.6 | 98.3 | 100.0 | 100.0 | 96.3 | 98.9 | 90.4 | 100.0 | 99.9 | 100.0 | 98.8 |
| DFFreq (Yan et al., 2026) | 100.0 | 98.7 | 98.4 | 100.0 | 100.0 | 98.1 | 99.8 | 89.0 | 87.5 | 100.0 | 99.9 | 98.7 | 99.5 | 99.7 | 100.0 | 99.3 | 99.9 | 98.1 |
| SAFE (Li et al., 2025b) | 100.0 | 99.5 | 97.8 | 99.9 | 100.0 | 96.6 | 100.0 | 73.8 | 91.0 | 100.0 | 100.0 | 98.1 | 99.9 | 99.9 | 100.0 | 99.9 | 100.0 | 97.4 |
| Effort (Yan et al., 2024b) | 100.0 | 100.0 | 100.0 | 99.3 | 99.2 | 100.0 | 100.0 | 91.1 | 97.9 | 100.0 | 100.0 | 97.5 | 99.8 | 98.9 | 100.0 | 100.0 | 99.9 | 99.0 |
| NS-Net (Yan et al., 2025b) | 100.0 | 100.0 | 99.7 | 99.8 | 99.9 | 97.0 | 100.0 | 99.5 | 95.2 | 100.0 | 100.0 | 100.0 | 99.8 | 97.0 | 100.0 | 100.0 | 100.0 | 99.2 |
| Ours | 100.0 | 100.0 | 99.9 | 100.0 | 100.0 | 98.8 | 100.0 | 99.8 | 98.3 | 100.0 | 100.0 | 99.8 | 100.0 | 99.6 | 100.0 | 100.0 | 100.0 | 99.8 |

semantic components from the image-task gradient. More importantly, different from existing methods (Zhang et al., 2025; Yan et al., 2025b) that directly discard all semantic information, our approach only modifies the gradient update direction to be orthogonal to the harmful semantic component. In this way, we retain beneficial semantic cues while simultaneously improving the model's generalization.

### 4.3. Prior Alignment

Similar to the approach described in Section 4.2, we define the negative direction of the image gradient as the *beneficial direction*, expressed as:

$$g_{\text{help}} \triangleq g_{\text{img}}^- = \left[ \nabla_{f^T} \mathcal{L}_{\text{img}}^T \right]_- \in \mathbb{R}^d, \tag{12}$$

which provides effective descent guidance toward downstream objectives while preserving the priors learned during pre-training.

To further maintain the transferable priors and geometric structures established by CLIP's pre-training, an intuitive solution is to directly constrain the update gradient with $g_{\text{help}}$. However, directly optimizing with $g_{\text{help}}$ as an explicit target is non-trivial. Thus, we introduce a linear alignment regularization term at the feature level:

$$\mathcal{L}_{\text{align}} = \langle f, g_{\text{help}} \rangle, \tag{13}$$

whose gradient with respect to $f$ is:

$$\nabla_f \mathcal{L}_{\text{align}} = g_{\text{help}}. \tag{14}$$

In essence, this mechanism injects beneficial descent directions derived from the pre-trained model into the fine-tuning process, thereby promoting lightweight alignment with transferable priors and enhancing the model's overall detection capability.

### 4.4. Overall Process and Loss Function

The overall training objective of DGS-Net is formulated as:

$$\mathcal{L} = \mathcal{L}_{\text{img}} + \mathcal{L}_{\text{text}} + \lambda \mathcal{L}_{\text{align}}, \tag{15}$$

where $\lambda$ is hyperparameter that balances the alignment term.

During backpropagation, the gradient propagated to the image feature $f$ is replaced by

$$g_{\text{final}} = \underbrace{\left( I - \hat{g}_{\text{harm}} \hat{g}_{\text{harm}}^\top \right) \nabla_f \mathcal{L}_{\text{img}}}_{\text{Orthogonal Suppression}(\bar{g})} + \underbrace{\lambda\, g_{\text{help}}}_{\text{Prior Alignment}}, \tag{16}$$

which jointly enforces semantic suppression and prior preservation. By the chain rule, the parameter update can be expressed as:

$$\theta \leftarrow \theta - \mu \left( \frac{\partial E_{\text{img}}(x; \theta)}{\partial \theta} \right)^\top g_{\text{final}}, \tag{17}$$

where $\mu$ denotes the learning rate. Through the integration of *Orthogonal Suppression* and *Prior Alignment*, the feature

*Table 3.* **Cross-model Accuracy (Acc.) Performance on the AIGIBench (Li et al., 2025c) Dataset.** The first row represents the accuracy of detecting real images (R.Acc.), and the others are the accuracy of detecting fake images (F.Acc.).

| Generator | CNN-Spot | UnivFD | FreqNet | NPR | Ladeda | AIDE | C2P-CLIP* | DFFreq | SAFE | Effort | NS-Net | Ours |
|---|---|---|---|---|---|---|---|---|---|---|---|---|
| Real Image | **98.2** | 73.3 | 64.6 | 93.8 | 91.7 | 88.1 | 93.8 | 91.8 | 92.4 | 96.9 | 88.3 | 83.0 |
| ProGAN | 95.3 | 98.9 | 99.4 | 98.9 | 99.7 | 95.3 | 100.0 | 99.3 | 99.7 | 100.0 | 100.0 | **100.0** |
| R3GAN | 2.3 | 94.1 | 59.9 | 8.4 | 19.5 | 99.0 | 69.0 | 78.4 | 93.0 | 94.4 | 98.5 | **99.9** |
| StyleGAN3 | 9.1 | 82.6 | 98.2 | 63.6 | 93.2 | 91.1 | 99.2 | 95.5 | 94.6 | 95.0 | 99.6 | **99.7** |
| StyleGAN-XL | 0.7 | 96.7 | 95.5 | 28.2 | 80.5 | 91.7 | **99.9** | 15.6 | 89.4 | 82.3 | 99.3 | 99.6 |
| StyleSwim | 6.9 | 98.1 | 97.1 | 77.7 | 97.3 | 82.0 | 99.8 | 99.8 | 99.9 | 97.6 | 99.8 | **99.9** |
| WFIR | 1.1 | 92.8 | **95.4** | 7.9 | 19.2 | 42.9 | 34.9 | 3.5 | 8.4 | 91.1 | 69.3 | 84.5 |
| BlendFace | 6.2 | 5.5 | 0.3 | 0.0 | 0.0 | 23.2 | 3.8 | 1.5 | 2.6 | 11.2 | 16.2 | **69.6** |
| E4S | 4.1 | **46.9** | 1.1 | 0.0 | 0.0 | 6.6 | 0.6 | 1.2 | 1.3 | 18.2 | 4.4 | 44.8 |
| FaceSwap | 1.4 | 27.3 | 6.2 | 0.0 | 0.0 | 14.3 | 4.6 | 7.4 | 3.0 | 14.1 | 11.1 | **66.6** |
| InSwap | 9.7 | 8.2 | 0.9 | 0.0 | 0.0 | 11.4 | 5.5 | 2.8 | 1.9 | 17.3 | 16.9 | **70.7** |
| SimSwap | 6.2 | 8.6 | 0.6 | 0.0 | 0.0 | 21.5 | 8.4 | 1.8 | 2.5 | 35.2 | 21.7 | **81.4** |
| DALLE-3 | 9.8 | **75.2** | 68.2 | 21.2 | 9.7 | 24.5 | 0.7 | 3.4 | 0.5 | 12.9 | 0.1 | 2.8 |
| FLUX1-dev | 16.3 | 86.6 | 92.4 | 97.2 | 99.3 | 90.0 | 41.2 | 96.1 | **99.5** | 26.8 | 96.2 | 98.7 |
| Midjourney-V6 | 5.8 | 80.6 | 83.6 | 53.8 | 83.4 | 79.8 | 67.3 | 95.8 | 94.1 | 70.4 | 94.8 | **97.7** |
| GLIDE | 4.6 | 75.2 | 79.7 | 70.3 | 81.8 | 98.4 | 70.7 | 86.9 | 89.2 | 81.0 | 96.6 | **98.6** |
| Imagen3 | 4.2 | 84.2 | 81.5 | 78.2 | 92.6 | 93.9 | 71.7 | 51.9 | 94.8 | 84.8 | **99.5** | 99.4 |
| SD3 | 13.3 | 90.6 | 88.1 | 89.7 | 99.0 | 99.3 | 83.7 | 92.1 | 94.4 | 86.9 | **99.9** | 99.8 |
| SDXL | 7.2 | 88.0 | 98.9 | 79.0 | 98.3 | 97.6 | 84.4 | 98.0 | 99.9 | 86.4 | 99.9 | **100.0** |
| BLIP | 56.5 | 92.1 | 100.0 | 99.9 | 100.0 | 100.0 | 100.0 | 100.0 | 100.0 | 100.0 | 100.0 | **100.0** |
| Infinite-ID | 1.1 | 93.8 | 92.7 | 34.6 | 32.2 | 97.5 | 36.5 | 93.9 | 99.2 | 43.3 | 92.2 | **99.8** |
| PhotoMaker | 1.7 | 65.2 | 88.6 | 3.6 | 66.7 | 97.5 | 23.5 | 99.6 | **99.7** | 23.2 | 41.9 | 96.6 |
| Instant-ID | 8.1 | 96.9 | 93.9 | 34.1 | 82.4 | 97.0 | 38.4 | 100.0 | **100.0** | 46.2 | 78.9 | 98.7 |
| IP-Adapter | 6.0 | 92.0 | 92.0 | 71.8 | 90.6 | 93.5 | 82.1 | 97.8 | 97.2 | 88.2 | 99.8 | **99.9** |
| SocialRF | 7.5 | **55.5** | 39.3 | 21.9 | 19.4 | 18.4 | 14.3 | 18.4 | 17.1 | 16.9 | 18.4 | 19.2 |
| CommunityAI | 5.4 | **51.2** | 12.2 | 8.2 | 9.0 | 9.3 | 5.2 | 9.2 | 9.1 | 5.2 | 8.9 | 9.2 |
| Average | 15.0 | 71.5 | 66.6 | 43.9 | 56.4 | 67.8 | 51.5 | 59.3 | 64.7 | 58.7 | 67.4 | **81.6** |

extractor $E_{img}$ not only mitigates harmful semantic interference but also retains transferable pre-training priors, thereby enhancing generalization in AI-generated image detection.

During training, we update only the LoRA parameters of the CLIP image encoder ($\theta$) and the two classification heads, $h_{img}$ and $h_{text}$, while keeping all other parameters frozen. At inference time, only the image encoder $E_{img}$ and the linear classification head $h_{img}$ are used for image detection, ensuring computational efficiency.

# 5. Experiments

## 5.1. Settings

**Training Datasets.** We adopt the **Setting-II** of AIGIBench (Li et al., 2025c), which includes 144k images generated by ProGAN (Karras et al., 2018) and SDv1.4 (Rombach et al., 2022), along with real images.

**Testing Datasets.** To comprehensively evaluate the effectiveness of our method, we use three benchmarks called UniversalFakeDetect (Ojha et al., 2023) (Section B of the Appendix), AIGCDetectBench (Zhong et al., 2023), and AIGIBench (Li et al., 2025c), which include Guided, GLIDE, LDM, DALLE, ProGAN, CycleGAN, BigGAN, StyleGAN, StyleGAN2, GauGAN, StarGAN, WFIR, Deep-

fake, SDv1.4, SDv1.5, ADM, Midjourney, Wukong, VQDM, DALLE2, R3GAN, StyleGAN3, StyleGAN-XL, StyleSwim, BlendFace, E4S, FaceSwap, InSwap, SimSwap, DALLE-3, FLUX1-dev, Imagen3, SD3, SDXL, BLIP, Infinite-ID, PhotoMaker, Instant-ID, IP-Adapter, CommunityAI and SocialRF.

**Implementation Details.** The model is trained with the Adam optimizer (Kingma, 2014), a learning rate of $1 \times 10^{-4}$, and a batch size of 32 for only 1 epoch. The ViT-L/14 model of CLIP is adopted as the pre-trained model (Ojha et al., 2023). In addition, we add Low-Rank Adaptation to fine-tune the CLIP's image encoder, following (Zanella & Ben Ayed, 2024). The hyperparameter $\lambda$ is set to 0.2. We use Patch Selection (Yan et al., 2025b) to resize the image to $224 \times 224$. Importantly, to ensure a fair comparison, we retrain all baseline methods [1] on the same training set. More details are shown in Section A of the Appendix.

## 5.2. Comparison with the State-of-the-Art

**Evaluation on AIGCDetectBench.** The results of accuracy (Acc.) and average precision (A.P.) are shown in Table 1 and

---

[1] When training C2P-CLIP (Tan et al., 2025), we replace the original text generation module (ClipCap) with BLIP, keeping all other components unchanged.

*Table 4.* **Ablation Study of our method.** The accuracy averaged on the AIGCDetectBench and AIGIBench datasets.

| Orthogonal Suppression | Prior Alignment | AIGCDetectBench | AIGIBench |
|:---:|:---:|:---:|:---:|
| ✗ | ✗ | 79.9 | 52.4 |
| ✗ | ✓ | $83.3_{+3.4}$ | $58.0_{+5.6}$ |
| ✓ | ✗ | $88.0_{+8.1}$ | $61.7_{+9.3}$ |
| ✓ | ✓ | $97.6_{+17.7}$ | $81.6_{+29.2}$ |

*Table 5.* **Robustness on JPEG Compression and Gaussian Blur of Our Method.** The accuracy averaged on AIGCDetectBench.

| Method | JPEG Compression | | | Gaussian Blur | | |
|---|---|---|---|---|---|---|
| | QF=95 | QF=85 | QF=75 | $\sigma = 0.5$ | $\sigma = 1.0$ | $\sigma = 1.5$ |
| CNN-Spot (Wang et al., 2020) | 59.0 | 59.9 | 60.5 | 62.4 | 63.2 | 63.3 |
| UnivFD(Ojha et al., 2023) | 76.4 | 74.2 | 73.8 | 80.2 | 76.8 | 76.3 |
| NPR (Tan et al., 2024b) | 72.8 | 70.7 | 69.2 | 80.6 | 79.6 | 81.0 |
| AIDE (Yan et al., 2024a) | 74.6 | 74.1 | 70.3 | 88.6 | 75.8 | 81.1 |
| SAFE (Li et al., 2025b) | 57.7 | 59.6 | 60.4 | 88.5 | 80.6 | 83.0 |
| Effort(Yan et al., 2024b) | 83.6 | 82.6 | 81.8 | 87.3 | 86.7 | 87.0 |
| NS-Net (Yan et al., 2025b) | 85.6 | 82.3 | 79.6 | 88.8 | 86.1 | 85.9 |
| Ours | **88.9** | **85.0** | **82.4** | **92.7** | **89.7** | **89.9** |

*Table 6.* **Robustness on JPEG Compression and Gaussian Blur of Our Method.** The accuracy averaged on AIGIBench.

| Method | JPEG Compression | | | Gaussian Blur | | |
|---|---|---|---|---|---|---|
| | QF=95 | QF=85 | QF=75 | $\sigma = 0.5$ | $\sigma = 1.0$ | $\sigma = 1.5$ |
| CNN-Spot (Wang et al., 2020) | 53.8 | 54.3 | 54.5 | 56.5 | 56.5 | 56.6 |
| UnivFD(Ojha et al., 2023) | 65.6 | 66.3 | 65.1 | 72.2 | 68.6 | 68.6 |
| NPR (Tan et al., 2024b) | 59.0 | 58.9 | 58.6 | 62.9 | 62.0 | 62.9 |
| AIDE (Yan et al., 2024a) | 61.0 | 59.6 | 55.9 | 75.5 | 63.4 | 65.2 |
| SAFE (Li et al., 2025b) | 51.3 | 47.9 | 47.8 | **75.5** | 67.6 | 69.3 |
| Effort(Yan et al., 2024b) | 67.6 | 66.9 | 64.9 | 71.9 | 72.5 | 71.9 |
| NS-Net (Yan et al., 2025b) | 68.9 | 67.1 | 64.9 | 72.8 | 70.3 | 70.2 |
| Ours | **69.4** | **67.8** | **65.7** | 75.4 | **72.9** | **73.1** |

*Table 7.* **Linear Probing Results of Frozen Backbone.** DGS-Net can better maintain its transferable representation geometry structure during fine-tuning.

| Method | AIGCDetectBench | | AIGIBench | |
|---|---|---|---|---|
| | Acc. | A.P. | Acc. | A.P. |
| CLIP | 72.7 | 91.9 | 71.5 | 75.7 |
| CLIP-LoRA | 82.4 | 98.5 | 52.4 | **85.6** |
| DGS-Net | **93.3** | **98.8** | **76.7** | 85.4 |

2 respectively. Overall, our method achieves 97.6% mAcc. and 99.8% mA.P. across 17 test subsets. Although inference remains identical to C2P-CLIP, our method improves 15.2% mAcc. and 1.0% mA.P., indicating a substantial improvement in CLIP-based detection. Furthermore, compared with the state-of-the-art NS-Net, our method achieves an additional 4.4% improvement in mAcc. Notably, on the challenging Deepfake dataset, it reaches 96.7% in accuracy, whereas existing methods have historically shown very low.

**Evaluation on AIGIBench.** We show the results of accuracy (Acc.) on the AIGIBench dataset in Table 3. The additional metric about A.P. is shown in the Section C of the Appendix. Our method achieves 81.6% accuracy, a 10.1% improvement over the state-of-the-art, UnivFD. Notably, for Deepfake datasets such as BlendFace, where existing detection methods almost completely fail, our method delivers a substantial gain with an average accuracy improvement of about 50%. Nevertheless, performance remains poor on certain datasets (DALLE-3, SocialRF, CommunityAI), similar to other methods. This may be attributed to unknown post-processing operations, remaining an open challenge.

### 5.3. Ablation Studies

To further assess the effectiveness of our network, we perform ablation studies on its core components. The corresponding results are shown in Table 4. Compared with the baseline, our method achieves 17.7% and 29.2% improvements in accuracy on the AIGCDetectBench and AIGIBench datasets, respectively. This demonstrates that our method substantially enhances CLIP's ability to extract more generalizable artifact representations for universal AI-generated image detection. Moreover, the inclusion of any single component yields measurable performance gains, further demonstrating the effectiveness of our approach in

addressing the challenges of catastrophic forgetting. More ablation studies are shown in Section D of the Appendix.

### 5.4. Robustness Evaluation

In real-world scenarios, images are inevitably affected by unknown disturbances during transmission and interaction. To investigate robustness under such conditions, we further evaluate the performance of different detection methods against a variety of disturbances, such as JPEG compression and Gaussian blur. As illustrated in Table 5 and 6, our method consistently outperforms competing approaches, maintaining relatively high accuracy. Notably, our method exhibits almost no degradation in detection performance under varying levels of Gaussian blur interference, further demonstrating its strong robustness.

### 5.5. Geometric Structure Preservation

To demonstrate that DGS-Net preserves the transferable geometric structure of the original CLIP manifold, we evaluate the intrinsic transferability of the learned visual representations through linear probing. Specifically, we freeze the trained backbones of original CLIP, CLIP-LoRA, and DGS-Net, and re-train an identical fresh FC head. As shown in Table 7, CLIP-LoRA drops sharply on AIGIBench (52.4%), even below frozen CLIP (71.5%), indicating the reduced transferability of the learned backbone features after naive fine-tuning. In contrast, DGS-Net remains substantially stronger (76.7% on AIGIBench) with the same fresh-head protocol, showing that its frozen representations are more transferable across unseen generators. This further validates that DGS-Net can better maintain its transferable representation geometry structure during fine-tuning. Whats's more, the performance gap between linear probing and the original

end-to-end evaluation is expected, as the newly initialized linear classifier trained on frozen features is more prone to overfitting to the limited training generators.

## 6. Conclusion

In this paper, we review and analyze existing strategies for fine-tuning CLIP in AI-generated image detection, and identify that catastrophic forgetting in current fine-tuning paradigms undermines the transferable priors of the pre-trained model. To address these challenges, we propose a novel Distillation-Guided Gradient Surgery Network (DGS-Net). Specifically, we orthogonally project the image gradients of the training network onto the complement of the harmful directions estimated from text gradients, thereby suppressing components irrelevant to the detection. In addition, we inject the coordinate-wise negative component of a frozen CLIP's image gradients as a lightweight alignment, preserving the transferable priors and geometric structure established during large-scale pre-training. Together, they strike a balance between suppressing irrelevance and preserving transferable priors, substantially enhancing the generalization performance of AI-generated image detection.

## Acknowledgment

This work was supported in part by the National Natural Science Foundation of China under grant U22B2062, 62172232, 62502215, the Natural Science Foundation of Jiangsu under Grants BK20250735, the General Program of Natural Science Research in Universities of Jiangsu under Grants 25KJB520027, and Jiangsu Provincial Science and Technology Major Project (No. BG2024042).

## Impact Statement

This paper presents work aimed at advancing the field of Security in Trustworthy Machine Learning. It investigates and proposes an AI-generated image detection approach that demonstrates potential for mitigating the malicious use of generative models and may produce a positive social impact.

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

# A. Compared Detectors

To provide a comprehensive benchmark for comparison, we introduce 11 existing state-of-the-art detection methods, including CNN-Spot (CVPR 2020) (Wang et al., 2020), UnivFD (CVPR 2023) (Ojha et al., 2023), FreqNet (AAAI 2024) (Tan et al., 2024a), NPR (CVPR 2024) (Tan et al., 2024b), Ladeda (arXiv 2024) (Cavia et al., 2024), AIDE (ICLR 2025) (Yan et al., 2024a), C2P-CLIP (AAAI 2025) (Tan et al., 2025), DFFreq (arXiv 2025) (Yan et al., 2026), SAFE (KDD 2025) (Li et al., 2025b), Effort (ICML 2025) (Yan et al., 2024b) and NS-Net (arxiv 2025) (Yan et al., 2025b), details are as follow:

**1) CNN-Spot** (CVPR 2020) (Wang et al., 2020). CNN-Spot employs convolutional neural networks to detect synthetic content by analyzing common spatial artifacts in AI-generated images. It captures hierarchical features directly from pixel data, enabling the detection of generation anomalies.

**2) UnivFD** (CVPR 2023) (Ojha et al., 2023). UnivFD demonstrates that CLIP can effectively extract artifacts from images. By training a classifier on these features, it achieves strong cross-generator generalization performance.

**3) FreqNet** (AAAI 2024) (Tan et al., 2024a). FreqNet isolates high-frequency components of images via an FFT-based high-pass filter and introduces a frequency-domain learning block. This block transforms intermediate feature maps using FFT, applies learnable magnitude and phase adjustments, and reconstructs them with iFFT, enabling direct optimization in the frequency domain.

**4) NPR** (CVPR 2024) (Tan et al., 2024b). NPR targets structural artifacts introduced by up-sampling layers in generative models. It transforms input images into NPR maps that capture signed intensity differences between each pixel and its neighbors, explicitly revealing local dependency patterns characteristic of synthetic up-sampling operations.

**5) Ladeda** (arxiv 2024) (Cavia et al., 2024). LaDeDa is a patch-level deepfake detector that partitions each input image into $9 \times 9$ pixel patches and processes them using a BagNet-style ResNet-50 variant with its receptive field constrained to the same $9 \times 9$ region. The model assigns a deepfake likelihood to each patch, and the final prediction is obtained by globally pooling the patch-level scores.

**6) AIDE** (ICLR 2025) (Yan et al., 2024a). AIDE combines low-level patch statistics with high-level semantics for AI-generated image detection. It employs two expert branches: a semantic branch, which leverages CLIP-ConvNeXt embeddings to detect content inconsistencies, and a patchwise branch, which selects patches by spectral energy and applies a lightweight CNN to capture fine-grained artifacts.

**7) C2P-CLIP** (AAAI 2025) (Tan et al., 2025). C2P-CLIP concludes that CLIP achieves classification by matching similar concepts rather than discerning true and false. Based on this conclusion, they propose category common prompts to fine-tune the image encoder by manually constructing category concepts combined with contrastive learning.

**8) DFFreq** (TIFS 2026) (Yan et al., 2026). DFFreq first utilizes a sliding window to restrict the attention mechanism to a local window, and reconstruct the features within the window to model the relationships between neighboring internal elements within the local region. Then, a dual frequency domain branch framework consisting of four frequency domain subbands of DWT and the phase part of FFT is designed to enrich the extraction of local forgery features from different perspectives.

**9) SAFE** (KDD 2025) (Li et al., 2025b). SAFE replaces conventional resizing with random cropping to better preserve high-frequency details, applies data augmentations such as Color-Jitter and RandomRotation to break correlations tied to color and layout, and introduces patch-level random masking to encourage the model to focus on localized regions where synthetic pixel correlations typically emerge.

**10) Effort** (ICML 2025) (Yan et al., 2024b). Effort find that a naively trained detector very quickly shortcuts to the seen fake patterns, collapsing the feature space into a low-ranked structure that limits expressivity and generalization. Thus, they decompose the feature space into two orthogonal subspaces, for preserving pre-trained knowledge while learning forgery.

**11) NS-Net** (arxiv 2025) (Yan et al., 2025b). NS-Net uses the feature homogeneity extracted by the text encoder to replace the semantic information of the features extracted by the image encoder, and uses NULL-Space to decouple the semantic information, retaining the artifact information related to the forgery detection task.

*Table 8.* **Cross-Diffusion-Sources Evaluation on the Diffusion Test Set of UniversalFakeDetect (Ojha et al., 2023).**

| Detection Method | Guided | | Glide_50_27 | | Glide_100_10 | | Glide_100_27 | | LDM_100 | | LDM_200 | | LDM_200_cfg | | DALLE | | Mean | |
|---|---|---|---|---|---|---|---|---|---|---|---|---|---|---|---|---|---|---|
| | Acc. | A.P. | Acc. | A.P. | Acc. | A.P. | Acc. | A.P. | Acc. | A.P. | Acc. | A.P. | Acc. | A.P. | Acc. | A.P. | mAcc. | mA.P. |
| CNN-Spot (Wang et al., 2020) | 51.3 | 70.8 | 52.2 | 76.9 | 52.2 | 78.0 | 52.4 | 76.8 | 60.6 | 88.7 | 62.0 | 90.1 | 68.0 | 93.8 | 51.6 | 62.7 | 56.3 | 79.7 |
| UnivFD (Ojha et al., 2023) | 64.2 | 81.0 | 84.0 | 92.5 | 84.5 | 92.1 | 84.3 | 92.3 | 82.5 | 91.4 | 82.5 | 91.4 | 73.4 | 84.7 | 69.0 | 80.3 | 78.1 | 88.2 |
| FreqNet (Tan et al., 2024a) | 68.2 | 78.5 | 89.8 | 97.8 | 90.2 | 97.9 | 88.8 | 97.6 | 97.6 | 99.7 | 98.1 | 99.8 | 97.3 | 99.7 | 87.6 | 95.5 | 87.6 | 95.5 |
| NPR (Tan et al., 2024b) | 64.6 | 83.0 | 80.9 | 99.3 | 80.2 | 99.0 | 79.9 | 99.3 | 88.0 | 99.9 | 86.9 | 99.9 | 94.6 | 100.0 | 63.6 | 88.9 | 79.8 | 97.4 |
| Ladeda (Cavia et al., 2024) | 79.8 | 87.5 | 98.3 | 99.8 | 98.2 | 99.8 | 98.5 | 99.8 | 98.6 | 99.9 | 98.5 | 99.9 | 98.0 | 99.8 | 83.9 | 98.5 | 94.2 | 98.1 |
| AIDE (Yan et al., 2024a) | 90.1 | 98.5 | 98.3 | 99.9 | 97.9 | 99.8 | 98.1 | 99.8 | 98.4 | 99.9 | 98.3 | 100.0 | 98.5 | 100.0 | 96.7 | 99.5 | 97.0 | 99.7 |
| C2P-CLIP* (Tan et al., 2025) | 80.3 | 98.6 | 94.3 | 99.6 | 93.0 | 99.8 | 89.3 | 99.4 | 100.0 | 100.0 | 100.0 | 100.0 | 100.0 | 100.0 | 100.0 | 100.0 | 94.6 | 99.7 |
| DFFreq (Yan et al., 2026) | 90.3 | 99.2 | 94.8 | 99.2 | 95.7 | 99.2 | 93.9 | 99.0 | 99.3 | 100.0 | 99.2 | 100.0 | 99.2 | 100.0 | 95.2 | 99.5 | 96.0 | 99.5 |
| SAFE (Li et al., 2025b) | 80.7 | 96.0 | 94.8 | 98.5 | 95.2 | 98.4 | 93.2 | 97.9 | 97.2 | 99.9 | 97.4 | 99.9 | 97.1 | 99.9 | 94.7 | 99.1 | 93.8 | 98.7 |
| Effort (Yan et al., 2024b) | 76.4 | 98.3 | 95.7 | 99.8 | 95.7 | 99.8 | 93.7 | 99.6 | 99.9 | 100.0 | 100.0 | 100.0 | 99.8 | 100.0 | 100.0 | 100.0 | 95.2 | 99.7 |
| NS-Net (Yan et al., 2025b) | 92.5 | 99.4 | 98.2 | 99.9 | 99.0 | 99.9 | 98.2 | 99.8 | **100.0** | 100.0 | **100.0** | 100.0 | **100.0** | 100.0 | **100.0** | 100.0 | 98.5 | 99.9 |
| Ours | **97.9** | **99.8** | **98.9** | **99.9** | **99.9** | **100.0** | **98.5** | **99.9** | 99.6 | **100.0** | 99.7 | **100.0** | 99.4 | **100.0** | 99.5 | **100.0** | **99.0** | **100.0** |

*Table 9.* **Cross-model Average Precision (A.P.) Performance on the AIGIBench (Li et al., 2025c) Dataset.**

| Generator | CNN-Spot | UnivFD | FreqNet | NPR | Ladeda | AIDE | C2P-CLIP* | DFFreq | SAFE | Effort | NS-Net | Ours |
|---|---|---|---|---|---|---|---|---|---|---|---|---|
| ProGAN | 99.9 | 99.9 | 100.0 | 100.0 | 100.0 | 99.6 | 100.0 | 100.0 | 100.0 | 100.0 | 100.0 | 100.0 |
| R3GAN | 52.4 | 91.2 | 55.8 | 61.1 | 72.6 | 97.1 | 91.3 | 90.6 | 99.2 | 98.2 | 91.3 | 97.1 |
| StyleGAN3 | 85.2 | 84.5 | 92.4 | 91.7 | 96.9 | 91.4 | 99.5 | 98.1 | 94.7 | 98.9 | 99.4 | 98.3 |
| StyleGAN-XL | 50.2 | 93.3 | 83.7 | 75.3 | 93.1 | 93.2 | 98.9 | 69.5 | 90.8 | 97.0 | 96.1 | 97.9 |
| StyleSwim | 76.2 | 95.2 | 91.6 | 94.9 | 98.5 | 89.3 | 99.6 | 99.5 | 96.7 | 99.1 | 97.8 | 99.6 |
| WFIR | 65.5 | 83.0 | 57.0 | 65.5 | 86.9 | 90.8 | 99.6 | 89.0 | 60.7 | 100.0 | 99.4 | 99.9 |
| BlendFace | 73.2 | 35.2 | 34.0 | 34.7 | 42.1 | 54.2 | 48.9 | 42.3 | 44.1 | 66.1 | 54.4 | 65.3 |
| E4S | 68.7 | 57.0 | 34.5 | 34.4 | 49.3 | 44.3 | 48.3 | 41.0 | 45.6 | 76.9 | 49.4 | 56.8 |
| FaceSwap | 58.3 | 52.3 | 42.9 | 43.6 | 40.9 | 56.3 | 70.4 | 64.9 | 69.6 | 85.8 | 62.5 | 79.0 |
| InSwap | 77.5 | 40.1 | 41.8 | 40.7 | 47.4 | 54.6 | 65.6 | 57.3 | 62.3 | 83.1 | 67.5 | 83.1 |
| SimSwap | 69.7 | 40.3 | 41.7 | 42.7 | 42.3 | 62.7 | 68.5 | 57.7 | 59.9 | 89.7 | 69.5 | 86.0 |
| DALLE-3 | 68.4 | 76.4 | 60.2 | 70.0 | 59.8 | 63.1 | 56.3 | 48.5 | 49.5 | 60.6 | 42.2 | 40.5 |
| FLUX1-dev | 72.1 | 79.5 | 87.0 | 99.0 | 98.7 | 93.4 | 84.3 | 98.2 | 99.0 | 74.4 | 94.1 | 96.8 |
| Midjourney-V6 | 59.8 | 61.5 | 55.9 | 76.9 | 86.9 | 83.0 | 84.6 | 94.2 | 94.5 | 90.0 | 90.0 | 92.8 |
| GLIDE | 59.7 | 80.2 | 76.5 | 94.3 | 95.0 | 97.7 | 94.9 | 97.7 | 96.1 | 96.5 | 96.1 | 97.3 |
| Imagen3 | 57.0 | 79.3 | 80.2 | 94.4 | 97.2 | 95.2 | 93.9 | 84.7 | 97.0 | 97.8 | 95.9 | 94.6 |
| SD3 | 72.7 | 87.2 | 81.9 | 97.2 | 98.7 | 98.3 | 97.1 | 97.0 | 99.3 | 97.9 | 98.9 | 99.1 |
| SDXL | 63.8 | 88.0 | 95.0 | 94.4 | 98.5 | 95.7 | 97.1 | 99.1 | 99.4 | 98.0 | 98.2 | 99.1 |
| BLIP | 92.8 | 95.8 | 99.8 | 100.0 | 99.9 | 99.5 | 100.0 | 100.0 | 100.0 | 100.0 | 100.0 | 100.0 |
| Infinite-ID | 49.1 | 89.6 | 73.6 | 80.4 | 76.9 | 94.7 | 84.2 | 97.4 | 98.5 | 85.7 | 91.8 | 97.8 |
| PhotoMaker | 57.8 | 72.2 | 74.2 | 53.4 | 90.7 | 95.6 | 69.7 | 99.0 | 96.9 | 69.2 | 67.8 | 94.4 |
| Instant-ID | 79.9 | 93.5 | 86.0 | 79.2 | 90.4 | 96.3 | 84.9 | 99.9 | 94.8 | 87.1 | 87.7 | 97.2 |
| IP-Adapter | 65.5 | 87.3 | 79.4 | 91.7 | 94.3 | 95.4 | 95.4 | 98.8 | 94.8 | 98.1 | 96.2 | 99.0 |
| SocialRF | 50.6 | 55.2 | 58.1 | 68.4 | 68.3 | 65.0 | 67.7 | 66.9 | 63.5 | 65.2 | 74.6 | 71.1 |
| CommunityAI | 56.7 | 73.7 | 63.5 | 62.9 | 56.3 | 61.0 | 53.2 | 64.2 | 64.3 | 64.3 | 54.0 | 52.1 |
| Average | 67.3 | 75.7 | 69.9 | 73.9 | 79.3 | 82.7 | 82.2 | 82.2 | 82.9 | 87.2 | 83.0 | **87.8** |

*Table 10.* **Cross-model Accuracy (Acc.) Performance on the AIGCDetectBench (Zhong et al., 2023) Dataset.** We further experimentally verify the beneficial and harmful components of the gradient.

| Components | Real Image | Generative Adversarial Networks | | | | | | | Other | | Diffusion Models | | | | | | | | mAcc. |
|---|---|---|---|---|---|---|---|---|---|---|---|---|---|---|---|---|---|---|---|
| | | Pro-GAN | Cycle-GAN | Big-GAN | Style-GAN | Style-GAN2 | Gau-GAN | Star-GAN | WFIR | Deep-fake | SDv1.4 | SDv1.5 | ADM | GLIDE | Mid-journey | Wukong | VQDM | DALLE2 | |
| - (Baseline) | **99.9** | 99.9 | 99.4 | 99.6 | 83.1 | 87.0 | 80.6 | 100.0 | 9.8 | 39.3 | 100.0 | 99.9 | 52.1 | 75.2 | 54.1 | 98.9 | 89.7 | 69.2 | 79.9 |
| $g_{img}$, $g_{text}^+$ | 99.5 | 99.9 | 99.7 | 99.7 | 79.7 | 76.0 | 90.9 | 100.0 | 4.8 | 32.7 | 100.0 | 99.9 | 61.6 | 85.7 | 48.8 | 99.6 | 93.7 | 95.8 | 81.6 |
| $g_{img}^+$, $g_{text}$ | 99.5 | 100.0 | 100.0 | 100.0 | 81.7 | 92.7 | 97.6 | 100.0 | 24.6 | 40.6 | 100.0 | 99.9 | 71.0 | 90.0 | 59.4 | 99.7 | 96.5 | 96.8 | 86.1 |
| $g_{img}$, $g_{text}$ | 98.1 | 99.9 | 100.0 | 100.0 | 91.2 | 96.9 | 99.7 | 100.0 | 69.8 | 85.7 | 100.0 | 99.9 | 86.5 | 92.1 | 77.6 | 99.9 | 98.6 | 98.3 | 94.1 |
| $g_{img}^-$, $g_{text}^+$ (Ours) | 95.3 | **100.0** | **100.0** | **100.0** | **99.5** | **99.8** | **100.0** | **100.0** | **84.6** | **96.7** | **100.0** | **99.9** | **95.2** | **99.0** | **88.0** | **100.0** | **99.7** | **99.6** | **97.6** |

# B. Evaluation on UniversalFakeDetect

Table 8 underscores our method's capability in detecting unknown Diffusion-based fake images. Specifically, it achieves mean Accuracy (Acc.) and mean Average Precision (A.P.) of 99.0% and 100.0%, respectively. Our method achieves nearly 100% detection accuracy on almost all datasets, with particularly strong performance on the Guided dataset, where it

surpasses the best existing method by 5.4%, demonstrating the method's strong generalization ability.

## C. AP Result on AIGIBench

Follow the evaluation metrics used in baselines (Ojha et al., 2023; Tan et al., 2024a), which include average precision score (A.P.) and accuracy (Acc.), we add the A.P. results on AIGIBench (Li et al., 2025c), as shown in Table 9. Furthermore, our method achieves state-of-the-art performance among the baselines in terms of AP, underscoring the superiority of the proposed approach. This indicates that our method not only excels in overall prediction accuracy but also maintains robust performance across different decision thresholds, thereby demonstrating its effectiveness in distinguishing between classes.

## D. More Confirmatory Studies

As discussed in Section 3 and Section 4, we define the positive half-space of text gradients as harmful directions that should be suppressed, while the negative half-space of image gradients corresponds to beneficial directions that preserve the prior knowledge of the pre-trained model. To further validate this definition and theoretical reasoning, we conducted an additional experiment in which we directly used text gradients as harmful directions and treated all image gradients as beneficial directions. The results of this experiment are shown in Table 10. These results clearly demonstrate the necessity of categorizing gradients into harmful and beneficial components and treating them separately. For instance, injecting all image gradients as beneficial yields only limited improvements over the baseline, particularly on diffusion models; likewise, removing all text gradients as harmful results in performance substantially below our approach. In contrast, our method selectively projects and removes only the positive half-space of text gradients (harmful semantics) while injecting only the negative half-space of image gradients (beneficial priors). This strategy achieves the highest mAcc of 97.6%, approaching saturation on multi-class GANs and producing the most significant gains on diffusion models, thereby substantially improving cross-generator generalization.

