# OpenReview forum: "DGS-Net: Distillation-Guided Gradient Surgery for CLIP Fine-Tuning in AI-Generated Image Detection"
_ICML.cc/2026/Conference — ICML 2026 spotlight_

### Official Review · Reviewer_RXWh · 2026-03-08

**Soundness:** 2
**Presentation:** 3
**Significance:** 3
**Originality:** 3
**Overall Recommendation:** 4
**Confidence:** 4

**Summary:**

DGS-Net addresses the problems of poor generalization and knowledge forgetting when fine-tuning CLIP for AIGC detection. The authors propose a new fine-tuning method that works in the gradient space. It uses the text branch to identify "harmful semantics." Then, it uses orthogonal projection to remove the parts that lead to overfitting. At the same time, it uses "beneficial gradients" from a frozen CLIP model to guide the updates. With only one round of training, this method stays stable across 50 different generative models. It provides a helpful approach for adapting large-scale multimodal pre-trained models to digital forensics.

**Compliance With Llm Reviewing Policy:**

Affirmed.

**Final Justification:**

DGS-Net addresses the problems of poor generalization and knowledge forgetting when fine-tuning CLIP for AIGC detection. The authors propose a new fine-tuning method that works in the gradient space. At first, my greatest doubt about this article stemmed from the rationality of the "harmful" and "beneficial" definition constraints in the gradient space. In the rebuttal section, the author addressed most of my questions, so I recommend WA.

**Key Questions For Authors:**

See  Weaknesses.  If the authors can effectively answer the reviewer's questions, the reviewer will consider giving more points.

**Limitations:**

See  Weaknesses

**Strengths And Weaknesses:**

### Strengths

1. The paper is well-organized with clear logic. The layout and terminology are accurate, professional, and follow academic standards.

2.  Compared to previous works using CLIP for deepfake detection, this study provides more detailed modeling. Splitting gradients into "positive and negative" components to guide the representation distribution is a novel approach. Additionally, converting the text branch into a "negative feedback signal" by leveraging CLIP's image-text alignment is an interesting innovation.

3.  The evaluation is extensive. Testing the method across a benchmark of 50 different generative models effectively demonstrates its robustness and generalizability.

4.  The model shows impressive efficiency, reaching SOTA performance across diverse datasets with only a single epoch of fine-tuning.

---

### Weaknesses

1. The core of DGS-Net is gradient splitting, which models "harmful" and "beneficial" information based on whether the loss increases or decreases. However, these labels are highly subjective; in optimization theory, a gradient is simply a derivative of the loss function. This relies on a strong mathematical assumption that the CLIP feature space is decoupled. While the authors use t-SNE for visualization, they do not provide a clear proof of the true correspondence between the regularization effect and the claimed "harmful/beneficial" directions. This leads to two major doubts: (1) whether the operation is essentially a form of directional regularization, where large-scale gradient masking passively improves generalization by preventing the model from fitting small, unstable correlations rather than precisely removing "semantics"; and (2) whether the concept of harmful/beneficial gradient splitting is merely an intuitive, post-hoc hypothesis reversed-engineered to justify the experimental results.

2.  t-SNE can only demonstrate cluster separability and local neighborhood relationships. The claim that "geometric structure" (line 50) are preserved is technically inappropriate for t-SNE. The authors should provide mathematical proof or more rigorous experimental analysis to justify the gradient splitting logic.

3. Practical deployment is a key requirement for deepfake detection. DGS-Net involves multiple forward and backward passes due to distillation and text branches. The absence of data regarding training/validation time and memory consumption makes it difficult for the community to evaluate the actual utility and cost of the model.

4. The code or weights are not open-source. Given the theoretical gaps and the complexity of the gradient operations, the lack of reproducible files makes it difficult to verify the reliability and feasibility of the reported experimental results.

---

> ### Author Rebuttal · Authors · 2026-03-26
>
> **Q1: On whether harmful/beneficial splitting is subjective, post-hoc, or just generic masking.**
>
> **1) Mathematical Grounding over Subjective Labeling.** We agree that a gradient is fundamentally a derivative. However, our classification of "harmful" and "beneficial" is strictly grounded in optimization theory, it is a local first-order optimization definition at each update step: coordinates with positive gradient locally increase loss under positive perturbation, while negative coordinates locally decrease it. We have formally derived these dynamics in Section 3.
>
> **2) DGS-Net as Structured Regularization:** We concur that our method can be viewed as a form of directional regularization. However, its objective is not the "precise global removal" of semantics, but rather the dynamic mitigation of catastrophic forgetting during fine-tuning. Unlike generic, stochastic gradient masking that passively prevents fitting to micro-noise, DGS-Net performs structured regularization: we use Prior Alignment to inject beneficial pre-trained gradients to preserve transferable knowledge, while simultaneously suppressing harmful semantic optimization directions via Orthogonal Projection. This combination is not a post-hoc hindsight assumption, but a mathematically derived constraint specifically to navigate the representation collapse.
>
> **3) Direct Experimental Validation.** To definitively prove that our "harmful/beneficial" splitting captures true feature correspondence rather than acting as random regularization, we conducted an ablation study on the gradient components (Table I below). In this experiment, we isolated the Orthogonal Suppression mechanism:
>
> - When we project the training gradients strictly onto our defined harmful semantic directions (line 2), performance drops catastrophically, nearly reverting to the baseline. This confirms that these specific directions are genuinely detrimental to forensic generalization.
>
> - When we blindly suppress the entire text gradient (line 3), performance also drops, proving that semantics contain beneficial components that shouldn't be fully masked. Further extensive validation experiments are provided in Table 9 in the Appendix.
>
> - These empirical results, both mathematically and practically, validate our analysis: DGS-Net performs a precise, directionally-aware surgery, not a passive noise-masking artifact.
>
> **Table I. Ablation for Orthogonal Suppression.**
>
> |Method|AIGCDetectBench|AIGIBench|
> |:-:|:-:|:-:|
> |-|79.9|52.4|
> |$g_{text-align}^{+}$|81.0|53.7|
> |$g_{text}$|85.7|56.5|
> |**$g_{text}^{+}$**|**88.0**|**61.7**|
>
> **Q2: Geometric structure preservation.**
>
> To demonstrate that DGS-Net preserves the transferable geometric structure of the original CLIP manifold, we evaluate the intrinsic transferability of our learned feature manifold. Specifically, we conducted a linear probing experiment: we froze the trained backbones of of original CLIP, CLIP-LoRA, and DGS-Net, and re-trained an identical fresh FC head. The results are shown in Table II.
>
> Specifically, CLIP-LoRA drops sharply on AIGIBench (52.4%), even below frozen CLIP (71.5%), indicating the reduced transferability of the learned backbone features after naive fine-tuning. In contrast, DGS-Net remains substantially stronger (76.7% on AIGIBench) with the same fresh-head protocol, showing that its frozen representations are more transferable across unseen generators. This further validates that DGS-Net can better maintain its transferable representation geometry  structure during fine-tuning. Table 9 in the Appendix and Table I above further validate the rationality and effectiveness of our gradient surgery.
>
> **Table II.Linear Probing Results (Frozen Backbone)**
>
> |Method|AIGCDetectBench(Acc/AP)|AIGIBench(Acc/AP)|
> |:-:|:-:|:-:|
> |CLIP|72.7/91.9|71.5/75.7|
> |CLIP-LoRA|82.4/98.5|52.4/85.1|
> |**DGS-Net**|**93.3/98.8**|**76.7/85.4**|
>
> **Q3: The problem of practical deployment.**
>
> As shown in Table III in our response to Reviewer umnY, we provide a comprehensive comparison of training and inference costs between our DGS-Net and the CLIP-LoRA baseline.
>
> **1) Training cost.** DGS-Net introduces a moderate offline training overhead due to the auxiliary text/teacher branches used for gradient surgery: +2.7 GB training VRAM (19.7 GB vs 17.0 GB) and +20 min training time (1h40m vs 1h20m).
>
> **2) Inference cost.** Importantly, these auxiliary branches are removed at test time. Therefore, deployment uses the same image encoder + classifier as CLIP-LoRA, with essentially the same inference footprint under the same test protocol. Given the substantial generalization gain (+29.2% mAcc on AIGIBench), we believe this additional offline training cost is practically acceptable.
>
> **Q4: Open Source Commitment.**
>
> We thank the reviewer for highlighting the importance of reproducibility. We explicitly commit to releasing the complete source code and trained model weights on GitHub upon the official publication of this paper.

---

> > ### Author Rebuttal · Reviewer_RXWh · 2026-04-01
> >
> > The author effectively addressed my concerns and I raised my rating from 3 to 4.

---

> > > ### Author Response · Authors · 2026-04-01
> > >
> > > We are deeply grateful for your thoughtful review and supportive assessment of our work. It is encouraging to know that our responses have resolved your initial reservations. We are committed to incorporating your final suggestions into the revised version, and we stand ready to provide any additional clarification you might need.

---

### Official Review · Reviewer_umnY · 2026-03-09

**Soundness:** 4
**Presentation:** 3
**Significance:** 3
**Originality:** 4
**Overall Recommendation:** 4
**Confidence:** 4

**Summary:**

This paper proposes a novel framework, DGS‑Net, aimed at improving the generalization performance of AI‑generated image detection by mitigating catastrophic forgetting in the fine‑tuning of CLIP. The authors note that standard fine‑tuning strategies(e.g., LoRA) often degrade the transferable priors learned during pre‑training. To tackle this issue, the core idea is to decompose gradients in the representation space into detrimental and beneficial components: text-branch gradients are utilized to suppress task-irrelevant semantic directions (Orthogonal Suppression), while frozen image-encoder gradients are employed to inject beneficial descent directions (Prior Alignment). Extensive experiments across 50 generative models demonstrate a significant average accuracy improvement of 6.6% over state-of-the-art methods.

**Compliance With Llm Reviewing Policy:**

Affirmed.

**Key Questions For Authors:**

1.This method depends on text descriptions generated by BLIP in the training phase. It would be valuable to clarify whether inaccurate or irrelevant text descriptions may influence system performance, and whether different text descriptions made by other models would lead to inconsistent experimental results.

2.This paper sets a fixed hyperparameter λ = 0.2 for the alignment loss but does not analyze its sensitivity. Such an analysis would improve the rigor of the paper.

3.The work defines the positive half-space of text gradients as harmful and the negative half-space of image gradients as beneficial, with empirical results supporting this design. Nevertheless, in complex loss landscapes, gradient directions may not be strictly binary. Are there gradient coordinates in a ambiguous gray-area? It is worth exploring whether such a simple binary partition may introduce limitations.

**Limitations:**

The authors adequately discuss limitations, noting poor performance on certain datasets (DALLE-3, SocialRF, CommunityAI) potentially due to unknown post-processing operations. They also acknowledge that semantic information has dual roles (partially beneficial, partially harmful) rather than being universally detrimental. However, they could strengthen the limitations section by discussing computational overhead of maintaining dual gradient streams during training.

**Strengths And Weaknesses:**

Strengths:

1.The paper demonstrates strong technical soundness. The idea of performing gradient-space decomposition to separate harmful and beneficial directions for AGI detection is clearly defined and mathematically motivated. The orthogonal projection formulation is elegant and computationally lightweight.

2.The paper is well-written, and the methodology is explained clearly, making it easy for the reader to grasp the core concepts and workflow.

3.The experimental design is comprehensive, and the evaluation is reasonable, covering 50 different generative models and including robustness tests against JPEG compression and Gaussian blur.


Weaknesses:

1.The paper is clearly written and the formulas are self-consistent. However, some intuitive explanations are slightly repetitive, and the theoretical motivation could be further strengthened.

2.The experiments are thorough, but all results are presented quantitatively. Including some visual examples would improve readability and presentation.

---

> ### Author Rebuttal · Authors · 2026-03-26
>
> **W1: Rephrase the motivation.**
>
> We thank the reviewer for this valuable suggestion.  We rephrase our motivation more concisely as follows:
>
> CLIP fine-tuning for AIGI detection suffers from catastrophic forgetting: it may improve in-domain separability but distort transferable pre-trained geometry, hurting unseen-generator generalization. To address this, DGS-Net performs selective gradient control during training:
>
> - Orthogonal Suppression removes harmful semantic-aligned components from task gradients;
> -  Prior Alignment injects lightweight beneficial directions from a frozen CLIP.
>
> In short, DGS-Net improves transferability by balancing task adaptation and prior preservation in gradient space.
>
> **W2: Strengthening the qualitative results.**
>
> We appreciate the reviewer's constructive suggestion to enhance our qualitative results. While our current Figure 1 provides a visual comparison between UnivFD (first row) and DGS-Net (third row), we fully agree that adding more diverse baseline visualizations would further clarify our method's superiority. In the revised version, we will add visualizations on Effort.
>
> **Q1: The impact of text generation models on performance.**
>
> We thank the reviewer for this insightful question. We clarify this issue from two aspects:
>
> **1) Influence of irrelevant text descriptions.** Our Orthogonal Suppression relies on the harmful semantic direction derived from the text gradient. If the generated text is inaccurate, the computed gradient direction may not correctly represent the distracting semantic priors, which can limit the model's ability to effectively suppress irrelevant variables.
>
> **2) Consistency across different text models.** We replaced BLIP with Qwen-1.8B for caption generation and retrained DGS-Net. On AIGCDetectBench, DGS-Net with Qwen-1.8B achieved 97.2% mAcc, which is close to the BLIP-based result (97.6% mAcc). This suggests that our method is not highly sensitive to a specific captioning model. We will include this cross-captioner comparison in the revised manuscript.
>
> **Q2: The hyperparameter experiment.**
>
> We thank the reviewer for this valuable suggestion.
>
> As shown in Table I, $\lambda=0.2$ gives the best overall performance. When $\lambda=0$, performance drops notably, showing that Prior Alignment is necessary. When $\lambda$ becomes too large (0.4 to 1.0), performance degrades, indicating that prior alignment should remain a lightweight alignment signal rather than a dominant objective. Excessive alignment may over-constrain feature adaptation and weaken artifact-specific learning.
>
> **Table I. The hyperparameter experiment.**
> |$\lambda$|AIGCDetectBench (Acc./A.P.)|AIGIBench (Acc./A.P.)|
> |:-:|:-:|:-:|
> |0.0|88.0/99.4|61.7/86.2|
> |**Ours (0.2)**|**97.6/99.8**|**81.6/87.8**|
> |0.4|94.2/99.8|75.0/87.5|
> |0.6|90.5/99.7|65.5/84.1|
> |1.0|84.8/99.2|55.8/86.1|
>
> **Q3: The limitations of a simple binary partition**
>
> We thank the reviewer for this insightful point.
>
> Our positive/negative partition is a local and pragmatic approximation based on first-order optimization signals at each training step, rather than a claim of globally valid harmful/beneficial decomposition. Therefore, we do not assume every coordinate is strictly “good” or “bad” across the whole training trajectory. Instead, DGS-Net uses this decomposition as a lightweight guidance mechanism to suppress likely harmful semantic components while preserving likely beneficial pre-trained priors, which is empirically validated by improved cross-generator generalization.
>
> We will clarify this limitation explicitly in the revised manuscript and discuss future extensions.
>
> **L1: Computational efficiency.**
>
> We thank the reviewer for this helpful suggestion. As shown in Table II, we provide a comprehensive comparison of training and inference costs between our DGS-Net and the CLIP-LoRA baseline.
>
> **1) Training phase.** Compared with CLIP-LoRA, DGS-Net introduces a moderate offline training overhead: about +2.7 GB VRAM (19.7 GB vs 17.0 GB) and +20 min (1h40m vs 1h20m), mainly due to the auxiliary text/teacher branches used for dual-gradient guidance.
>
> **2) Inference phase.** Importantly, these auxiliary branches are removed at test time. Therefore, under the same evaluation protocol, deployment uses only the LoRA-adapted image encoder and keeps essentially the same inference footprint/latency as the baseline. Given the substantial generalization gain (+29.2% mAcc on AIGIBench), we believe this additional offline training cost is a highly justifiable trade-off for real-world forensic applications.
>
> We commit to adding this comprehensive efficiency analysis to the Limitations section in our revised manuscript.
>
> **Table II. Comparison of computational efficiency.**
> |Method|Batch Size|Train VRAM|Trainable Params|Train Time|Inference VRAM|Inference Time (AIGIBench)|mAcc.|
> |:-|:-:|:-:|:-:|:-:|:-:|:-:|:-:|
> |CLIP-LoRA|32|17.0GB|885505|1h15min|5169MB|43min|52.4|
> |DGS-Net|32|19.7GB|886247|1h40min|5418MB|46min|81.6|

---

> > ### Author Rebuttal · Reviewer_umnY · 2026-04-01
> >
> > The authors provided a thorough rebuttal addressing most technical concerns, including hyperparameter sensitivity, cross-captioner robustness, and computational cost.

---

> > > ### Author Response · Authors · 2026-04-01
> > >
> > > We sincerely thank you for your constructive comments and valuable suggestions. We are pleased that our response addressed your main concerns. We will incorporate the suggested changes into the final version as recommended.
> > >
> > > If you have any further questions, we would be happy to provide further explanation.
> > >
> > > What's more, if our revisions have satisfactorily resolved your issues, we would greatly appreciate it if you could consider increasing your score.

---

### Official Review · Reviewer_45iJ · 2026-03-10

**Soundness:** 4
**Presentation:** 4
**Significance:** 4
**Originality:** 4
**Overall Recommendation:** 6
**Confidence:** 5

**Summary:**

This submission proposes DGS-Net (Distillation-Guided Gradient Surgery Network) for AI-generated image detection with CLIP fine-tuning. DGS-Net introduces a gradient-space decomposition into “harmful” vs “beneficial” directions, and combines two mechanisms: (i) Orthogonal Suppression, which projects image-task gradients to be orthogonal to the “harmful” semantic directions estimated from a text branch (text descriptions generated by BLIP); and (ii) Prior Alignment, which injects “beneficial” descent directions distilled from a frozen CLIP image encoder. Experiments across AIGCDetectBench, AIGIBench, and UniversalFakeDetect report strong improvements over prior detectors.

**Compliance With Llm Reviewing Policy:**

Affirmed.

**Final Justification:**

Overall, I am satisfied with the authors’ rebuttal and continue to find the paper interesting, sufficiently novel, and supported by thorough experimental validation. I am therefore raising my score to reflect this assessment.

**Key Questions For Authors:**

see weaknesses

**Limitations:**

see weaknesses

**Strengths And Weaknesses:**

Strengths

•	The proposed “gradient surgery + distillation” viewpoint is interesting and, if clarified, could be a useful recipe for CLIP tuning.
•	Extensive benchmark coverage (many generators) and strong reported numbers; ablations suggest both Orthogonal Suppression and Prior Alignment contribute.

Weaknesses

1.	A central motivation is that fine-tuning collapses the representation geometry and hurts generalization (Fig. 1). However, Fig. 1 also seems to show that even when real/fake clusters are well separated, generalization can still fail. This raises the possibility that the generalization drop is not primarily due to representation learning/geometry, but instead due to overfitting or miscalibration in the classifier head (or the way the head is trained/regularized), while the backbone features remain reasonably transferable. Concretely, the paper would be stronger if it could rule out: 1) “features are fine; the FC head is overfitting to training generators / thresholds,” versus 2) “features truly become less transferable due to geometry collapse.”

2.	The method introduces extra branches and gradient operations (text branch gradients; frozen teacher gradients; projection). The paper currently does not provide a clear compute/memory/time accounting vs plain LoRA fine-tuning, vs representative KD/regularization baselines.

3.	Fig. 3 experimental protocol is unclear (reproducibility + interpretation)

4.	Several equations are confusing and may be technically incorrect or at least easy to misread:

1)	Eq. (11): symbol/notation abuse; please ensure consistent usage and define all symbols precisely.

2)	Eq. (10): \Delta f is presented as an optimization variable, but it reads like an operation on features, while the text describes a gradient update direction. Consider rewriting in standard optimization notation and consistently distinguish feature perturbations from gradient directions.

3)	Eq. (13): L_{\text{align}}=\langle f, g_{\text{help}}\rangle implies you are pushing features f to align with a gradient vector from the teacher branch. This is unusual and needs stronger explanation. Intuitively one would align feature-to-feature (student vs teacher features), or gradient-to-gradient (student gradient vs teacher gradient). If the intended effect is to inject a descent direction, please provide a clearer derivation and interpretation.

5.	You already report real accuracy and per-generator fake accuracy (Table 1), which is useful. However, it is still important to report overall Accuracy (and/or balanced accuracy), in addition to R.Acc and F.Acc breakdown. Many readers expect an overall scalar metric for comparison. In addition, Please clarify whether you are following the original AIGCDetectBench evaluation setting strictly. If not, explain why and how it differs. Lastly, For some strong baselines (notably PatchCraft and SDAIE) in AIGCDetectBench, at least provide an apples-to-apples comparison.

6.	Tables 5/6 include only a subset of baselines. The selection criteria are unclear. In particular, Effort seems missing, even though earlier tables show it performing competitively (often top-2/top-3). Please explain: 1) why specific methods were included/excluded, 2) whether missing methods were infeasible to run, or had incompatible preprocessing, etc.

7.	The method depends on text descriptions (BLIP) to estimate semantic gradients; robustness to caption noise, domain mismatch, or adversarial captions is unclear.


The idea is promising and the empirical results are strong, but I am not fully convinced by the current justification of the core motivation, and several key protocol/notation issues reduce confidence. If the authors address the concerns above with clearer experimental isolation, improved reproducibility details, stronger related-work positioning, and a careful revision of the math/notation, I would likely update my score upward to 5: accept.

---

> ### Author Rebuttal · Authors · 2026-03-26
>
> **Q1: Representation geometry.**
>
> We conducted a Linear Probing experiment: we extracted the trained backbones of CLIP-LoRA and DGS-Net, froze them, and trained a fresh, identical FC head under the same protocol. The results are shown in Table II in our response to Reviewer RXWh.
>
>  - Evidence of Geometry Collapse. The newly trained FC head of CLIP-LoRA fails catastrophically on AIGIBench (52.4%), even performing significantly worse than the original CLIP (71.5%). This confirms that the backbone features themselves have become non-transferable. The overfitting seen in the head is merely a symptom of the underlying geometry collapse; the features have been shattered to fit training artifacts, leaving no generalizable forensic signal for a new head to leverage.
>
> - Verification of Transferability in DGS-Net. The fresh FC head of DGS-Net maintains superior generalization (76.7% on AIGIBench). This proves that DGS-Net extracts truly transferable forensic features by preserving the intrinsic geometry of the pre-trained space.
>
> These results definitively rule out "head-only overfitting." The generalization drop in standard fine-tuning is primarily due to representation collapse. DGS-Net’s gradient surgery successfully mitigates this collapse, ensuring that the learned forensic features remain valid even across diverse, unseen generators.
>
> **Q2: Computational costs and deployment overhead.**
>
> As shown in Table II of our response to Reviewer umnY, compared with CLIP-LoRA, DGS-Net adds moderate training-only overhead: training VRAM increases 2.7GB, and training time increases 25min. However, the extra branches are removed at inference. Therefore, deployment uses the same network as CLIP-LoRA, with essentially the same practical inference pipeline. Given the substantial generalization gain (+29.2% mAcc on AIGIBench), this additional offline training overhead is a reasonable trade-off for deployment.
>
> **Q3: Clarification of Fig. 3.**
>
> Specifically, we convert training images (ProGAN + SDv1.4) into captions using BLIP, extract text features with a frozen CLIP text encoder, and train a linear classifier, then directly evaluate it on cross-domain test sets. The mean accuracy is approximately 60%.
>
> This result indicates that semantic text cues are only partially correlated with real/fake labels: they contain limited transferable signals but are insufficient for robust generalization by themselves. Therefore, we selectively suppress harmful semantic directions, rather than fully relying on or fully removing semantics.
>
> **Q4: Formula modification.**
>
> **1) Eq.(10) & Eq.(11).** We acknowledge that using $\Delta f$ is potentially confusing. In the revision, we will replace it with $g^\*$ to explicitly denote an optimized gradient update direction.
>
> **2) Eq.(13).** We clarify that our goal is Directional Distillation: we want the teacher-derived descent direction to directly steer feature optimization during backpropagation. Since directly optimizing with $g_{help}$ as an explicit target is non-trivial, we define $L_{align}$ as Eq.(13). When $L_{align}$ is incorporated into the total objective (Eq.15), its gradient with respect to the student feature $f$ is exactly $g_{help}$, as derived in Eq.(14). This formulation ensures that the teacher signal $g_{help}$ is naturally injected into the student’s feature update direction. In the revised version, we will make the mathematical derivation more explicit.
>
> **Q5: Experimental comparison and protocol clarification**:
>
> **1) Overall metric reporting and evaluation setting.** We report balanced accuracy in Table I. We use the AIGIBench's training setup (ProGAN + SDv1.4) , retrain all compared methods for fairness, and follow the AIGCDetectBench test protocol.
>
> **2) Additional strong baselines.** We have added PatchCraft and SDAIE results (Table II). Both methods are retrained with the same training protocol as other baselines, DGS-Net still achieves the best generalization performance.
>
> **Table I. Cross-model bAcc. performance on the AIGCDetectBench.**
> |Method|bAcc.|
> |:-:|:-:|
> |UnivFD|81.92|
> |NPR|74.84|
> |AIDE|89.08|
> |C2P-CLIP*|90.59|
> |DFFreq|91.22|
> |SAFE|89.46|
> |Effort|93.61|
> |NS-Net|95.32|
> |**Ours**|**96.52**|
>
> **Table II. Cross-model performance on the AIGCDetectBench.**
> |Method|mAcc.|A.P|
> |:-:|:-:|:-:|
> |PatchCraft|70.0|94.3|
> |SDAIE|91.4|96.6|
> |**Ours**|**97.6**|**99.8**|
>
> **Q6: Supplement to robustness experiments.**
>
> We exclued some methods that were less robust. Effort's omission was an oversight. We have now added Effort and others in Tables II/III in our response to Reviewer ve6A. DGS-Net remains the top-performing method. We will include Effort in the revised manuscript.
>
> **Q7: Robustness to caption noise/domain mismatch**
>
> We acknowledge that severe caption noise, domain mismatch, or adversarial captions may still affect gradient quality. We will add this as an explicit limitation and we will conduct further exploration of this in subsequent research.

---

> > ### Author Rebuttal · Reviewer_45iJ · 2026-04-01
> >
> > Overall, I am satisfied with the authors’ rebuttal and continue to find the paper interesting, sufficiently novel, and supported by thorough experimental validation. I am therefore raising my score to reflect this assessment.
> >
> > Specifically, I appreciate:
> >
> > 1）the addition of the linear-probing control, which is helpful for better isolating representation transferability by freezing the backbone and retraining an identical head;
> >
> > 2）the inclusion of computational cost statistics, including memory usage and runtime, as well as the clarification that the additional branches are used only during training, which largely addresses concerns regarding deployment feasibility;
> >
> > 3）the planned revisions to the mathematical notation in Eqs. (10), (11), and (13), together with the clearer explanation of directional guidance, which resolves my earlier confusion; and
> >
> > 4）the inclusion of balanced accuracy, stronger baselines such as PatchCraft and SDAIE, and expanded robustness results, which improve the fairness and comparability of the evaluation.
> >
> > I encourage the authors to ensure that all rebuttal clarifications and additional analyses are fully incorporated into the revised manuscript.
> >
> > With regard to R. SBWN, I agree with the authors’ explanation concerning performance saturation and do not consider this issue to materially affect the paper’s suitability for acceptance. The motivation of the work is clear and compelling, and the paper provides useful insight into how foundation models can be more effectively utilized for AIGC detection.
> > I am also satisfied by the authors’ response to R. RXWh regarding gradient splitting, and I encourage them to incorporate this explanation into the revised manuscript. With these additions, I find the work technically solid, experimentally thorough, and likely to be of interest to the community. I therefore support its acceptance.

---

> > > ### Author Response · Authors · 2026-04-01
> > >
> > > We sincerely thanks for your thoughtful feedback and positive recommendation. We are pleased to hear that our revisions addressed your concerns. We will incorporate the suggested changes into the final version as recommended. We remain available to address any remaining questions or concerns you may have.

---

### Official Review · Reviewer_ve6A · 2026-03-14

**Soundness:** 3
**Presentation:** 4
**Significance:** 3
**Originality:** 3
**Overall Recommendation:** 5
**Confidence:** 4

**Summary:**

This paper proposes to use multimodal foundation models such as CLIP to detect AI-generated images while mitigating catastrophic forgetting and cutting off harmful priors when fine-tuning them for this task. To do so, the method introduces a framework to decompose the gradient into beneficial and harmful directions in order to preserve only the beneficial VLM priors. During training, the gradient relative to the image is projected onto the subspace complementary to the harmful directions obtained from text gradients to avoid keeping harmful semantic priors from the textual backbone, and is also aligned with the beneficial component from the image gradient in order to preserve task beneficial priors.

**Compliance With Llm Reviewing Policy:**

Affirmed.

**Final Justification:**

A technically solid paper which provide both intuitions and empirical justification of the method. The gains are strong and consistent over most benchmarks.

**Key Questions For Authors:**

see weaknesses

**Limitations:**

Failure cases could be added and discussed

**Strengths And Weaknesses:**

## Strenghts

* The paper is very well written and easy to read.

* Overall, while being simple, the method provides strong and consistent gains for AI content detection.

## Weaknesses

* The methods shows ``only'' 83\% accuracy on real images. I wonder if it does not trade off real image accuracy for fake image recall. What would be ROC or AUC here?

*  I wonder why the authors chose to change the text generation module of C2P-CLIP?

* Robustness results are interesting but somewhat limited, with only Gaussian blur and JPEG compression evaluation. How would the method behave against stronger stress tests: resizing, color transforms, watermarking, cropping, screenshots....

---

> ### Author Rebuttal · Authors · 2026-03-25
>
> **Q1: Additional metrics.**
>
> We appreciate the reviewer’s feedback.
>
> **1) Real image accuracy.** While our method experiences a partial decrease in Real Accuracy, this yields a significant gain in Fake Accuracy (F.Acc) across diverse generators, effectively preventing the model from missing highly realistic, unseen fakes in open-world scenarios. This overall superiority is quantitatively supported by the Average Precision (AP) reported in Tables 5, 7, and 8, where DGS-Net consistently outperforms all baselines, demonstrating a superior and more robust precision-recall balance.
>
> **2) AUC analysis.** To provide a more comprehensive view of our performance, we have computed the AUC as requested.  We selected a representative portion of the data and presented the mean AUC on AIGIBench. As shown in Table I, DGS-Net achieves the highest Mean AUC (88.8%) on the challenging AIGIBench dataset, outperforming strong baselines like NS-Net (85.1%) and DFFreq (85.6%). We will include all the results in the revised version.
>
> **Table I. Cross-model AUC Performance on the AIGIBench Dataset.**
> |Method|R3GAN|WFIR|BlendFace|FLUX1-dev|SDXL|PhotoMaker|mAUC(ALL)|
> |:-:|:-:|:-:|:-:|:-:|:-:|:-:|:-:|
> |UnivFD|92.3|82.3|23.8|86.0|91.4|77.3|77.5|
> |AIDE|97.9|87.8|60.4|93.4|96.9|94.9|84.0|
> |DFFreq|94.8|48.8|56.8|98.2|99.1|**96.3**|85.6|
> |SAFE|97.4|49.9|52.0|**99.5**|99.1|91.2|82.9|
> |NS-Net|95.6|99.5|65.8|95.6|98.5|69.4|85.1|
> |Effort|94.4|90.4|47.3|72.3|94.3|55.5|80.5|
> |**Ours**|**98.0**|**99.8**|**74.1**|97.4|**99.3**|95.2|**88.8**|
>
> **Q2: Reason for replacing the ClipCap.**
>
> We appreciate the reviewer's inquiry regarding our architectural choice.
>
> **1) Mitigating Image Encoder Bias.** ClipCap intrinsically relies on CLIP's image encoder for feature extraction to generate captions. This creates a potential "semantic circularity" where any pre-existing bias or representational collapse in the CLIP image encoder is propagated into the generated text. BLIP helps mitigate this bias to some extent.
>
> **2) BLIP provides stronger captions:** BLIP is widely recognized in the research community for its superior captioning quality compared to earlier models like ClipCap. It provides a stronger and more reliable semantic representation. By replacing this module, we were able to achieve a stronger and more robust version of the C2P-CLIP for comparison.
>
> **3) Fair and unified comparison protocol.** We replaced ClipCap with BLIP to standardize the text-generation pipeline across methods in our experiments, reducing confounding factors from different captioning modules.
>
> **Q3: More robustness experiments.**
>
> We appreciate the reviewers' suggestions and have further supplemented our robustness experiments, including Resize($2\times2 $, $4 \times4$) and color transforms(Brightness, Contrast). The results are shown in Table II and III, and our method still maintains the best robustness.
>
> **Table II. Robustness of Our Method. The accuracy averaged on AIGCDetectBench.**
>
> |Method|QF95|QF85|QF75|Blur0.5|Blur1.0|Blur1.5|R2x2|R4x4|Bri0.2|Con0.2|
> |:-:|:-:|:-:|:-:|:-:|:-:|:-:|:-:|:-:|:-:|:-:|
> |CNN-Spot|59.0|59.9|60.5|62.4|63.2|63.3|59.0|52.5|57.5|58.8|
> |UnivFD|76.4|74.2|73.8|80.2|76.8|76.3|78.3|72.3|80.1|81.0|
> |NPR|72.8|70.7|69.2|80.6|79.6|81.0|67.5|66.7|74.6|77.5|
> |AIDE|74.6|74.1|70.3|88.6|75.8|81.1|50.1|56.0|83.9|88.5|
> |DFFreq|73.1|69.6|67.1|92.0|88.4|87.9|59.4|56.0|86.2|89.9|
> |SAFE|57.7|59.6|60.4|88.5|80.6|83.0|57.4|50.0|90.7|91.3|
> |NS-Net|85.6|82.3|79.6|88.8|86.1|85.9|88.0|70.6|88.8|89.3|
> |Effort|83.6|82.6|81.8|87.3|86.7|87.0|84.9|80.4|89.1|88.7|
> |**Ours**|**88.9**|**85.0**|**82.4**|**92.7**|**89.7**|**89.9**|**88.3**|**86.5**|**92.4**|**92.9**|
>
> **Table III. Robustness of Our Method. The accuracy averaged on AIGIBench.**
>
> |Method|QF95|QF85|QF75|Blur0.5|Blur1.0|Blur1.5|R2x2|R4x4|Bri0.2|Con0.2|
> |:-:|:-:|:-:|:-:|:-:|:-:|:-:|:-:|:-:|:-:|:-:|
> |CNN-Spot|53.8|54.3|54.5|56.5|56.5|56.6|54.2|51.3|53.0|53.4|
> |UnivFD|65.6|66.3|65.1|72.2|68.6|68.6|71.5|68.6|71.9|72.6|
> |NPR|59.0|58.9|58.6|62.9|62.0|62.9|61.7|61.8|61.6|63.4|
> |AIDE|61.0|59.6|55.9|75.5|63.4|65.2|50.8|56.7|72.3|73.1|
> |DFFreq|63.9|61.2|58.1|75.5|63.4|65.2|54.1|54.0|71.8|74.2|
> |SAFE|51.3|47.9|47.8|**75.5**|67.6|69.3|50.7|50.3|**76.7**|76.0|
> |NS-Net|68.9|67.1|64.9|72.8|70.3|70.2|74.2|65.1|73.4|73.9|
> |Effort|67.6|66.9|64.9|71.9|72.5|71.9|70.4|69.0|73.0|72.3|
> |**Ours**|**69.4**|**67.8**|**65.7**|75.4|**72.9**|**73.1**|**75.0**|**74.1**|76.0|**76.4**|
>
> **L1: Failure case discussion.**
>
> We thank the reviewer for the suggestion.
>
> We observe clear failure cases on subsets such as DALLE-3, SocialRF, and CommunityAI, where all methods (including ours) degrade significantly. These cases likely involve stronger domain shifts and unknown post-processing pipelines, making artifact cues less stable and harder to transfer.
>
> In the revision, we will analyze likely error modes (e.g., heavy post-processing or social-platform compression), and discuss future directions such as robustness-oriented augmentation.

---

### Official Review · Reviewer_SBWN · 2026-03-24

**Soundness:** 3
**Presentation:** 2
**Significance:** 2
**Originality:** 2
**Overall Recommendation:** 3
**Confidence:** 4

**Summary:**

This paper proposes the Distillation-guided Gradient Surgery Network (DGS-Net) to address the issue of catastrophic forgetting when fine-tuning large pre-trained models (such as CLIP) for AI-generated image detection. The authors introduce a gradient-space decomposition method that separates harmful and beneficial descent directions during optimization. By projecting task gradients onto the orthogonal complement of harmful directions while aligning with beneficial priors distilled from a frozen CLIP encoder, the method aims to preserve transferable knowledge and suppress irrelevant features. Extensive experiments conducted on 50 generative models demonstrate that DGS-Net outperforms existing state-of-the-art methods by an average margin of 6.6%, achieving superior detection performance and generalization.

**Compliance With Llm Reviewing Policy:**

Affirmed.

**Key Questions For Authors:**

1.  **Regarding the experimental metrics:** A significant number of the evaluation results approach 100% accuracy. Given that standard benchmarks typically involve statistical fluctuations and redundancy, could the authors explain why the task appears to be saturated? Does the near-perfect performance indicate that the current evaluation protocol is no longer challenging enough to reflect true generalization capabilities?

2.  **Regarding the practical value:** The mathematical formulation of the gradient surgery is complex. Could the authors provide a straightforward, high-level explanation of the practical value of this method? Under what specific real-world scenarios (e.g., low-quality images, unseen generators) does this method provide the most significant advantage over a simple fine-tuned CLIP model?

3.  **Regarding Figure 1 visualization:** The visualization in Figure 1 appears to show that CLIP-LoRA results in concentrated red regions, while DGS-Net results in dispersed red regions. Could the authors clarify why the dispersed pattern represents better prior preservation? In the context of detecting AI-generated images, is clustering features not typically considered beneficial for discrimination?

**Limitations:**

Did not see any discussion. Please point out if there is any

**Strengths And Weaknesses:**

**Strengths:**

1.  **Comprehensive Experimental Validation:** The paper presents extensive experiments comparing a wide range of similar methods. The inclusion of multi-dimensional visual analyses makes the study thorough and well-structured, reflecting a solid engineering effort.

**Weaknesses:**

2.  **Clarity of Exposition:** While there are no significant grammatical errors, the writing suffers from a lack of clarity. The motivation behind the problem is not articulated clearly, and the description of the proposed method (DGS-Net) is overly complex and difficult to follow. Consequently, the specific benefits brought by the method are not well communicated to the reader.

3.  **Questionable Experimental Metrics:** The experimental results raise concerns. Many of the reported metrics approach 100% or perfect scores. While the proposed method shows marginal improvements, the lack of statistical variance or randomness in the results is concerning. If the task is saturated to the point where multiple methods achieve near-perfect scores, it suggests that the current evaluation benchmark may be outdated or insufficient to meaningfully distinguish between model capabilities.

4.  **Visualization Discrepancy:** In Figure 1, the visualization results for CLIP-LoRA appear qualitatively superior or more desirable compared to the proposed DGS-Net. The key distinction in the figure seems to be whether the red regions (presumably indicating artifacts or focus areas) are clustered or dispersed. However, the paper does not sufficiently explain why the proposed dispersion pattern is preferable to the clustering pattern exhibited by CLIP-LoRA.

---

> ### Author Rebuttal · Authors · 2026-03-25
>
> **Q1 & W2: Concerns regarding performance saturation, lack of statistical variance.**
>
> **1) Statistical variance.** Regarding the very low variance, our method is inherently stable across runs: we freeze the pre-trained CLIP backbone and only optimize lightweight modules, which substantially reduces optimization randomness.
>
> **2) Performance saturation.** We agree that near-100% performance on some older subsets (e.g., early GAN-based data) suggests those specific low-level, generator-artifact cues are largely solved. However, on more recent and challenging settings, especially newer generation pipelines (e.g., Midjourney, E4S), all methods show clear performance degradation, indicating that cross-model generalization remains an open problem.
>
> **3) Evaluation.** To evaluate this broader and more realistic setting, we include experiments on AIGIBench, a recent comprehensive benchmark. Our method achieves 81.6% accuracy, outperforming the strongest baseline (UnivFD) by 10.1%. We believe these results support our main claim: while some legacy subsets are saturated, the overall evaluation protocol remains challenging.
>
> **Q2 & W1: Motivation and practical value.**
>
> **1) The limitation of simple CLIP fine-tuning.** Pre-trained models like CLIP offer rich transferable representations, but standard fine-tuning (e.g., LoRA) can induce catastrophic forgetting and representation collapse, significantly degrading generalization across unseen generators.
>
> **2) Motivation for DGS-Net.** Raw pre-trained priors are not universally beneficial for detection (Fig. 1, Top Row), and semantic information can act as both beneficial guidance and harmful interference (Fig. 3). Thus, the core of DGS-Net is a selective gradient steering mechanism. We perform gradient surgery during training to identify and align with "beneficial gradients" from the pre-trained space to reinforce prior preservation. Simultaneously, we use orthogonal gradient projection to suppress the "harmful semantic components".
>
> **3) Practical value and real-world impact.** DGS-Net strikes a precision balance between suppressing irrelevance and preserving transferable priors, substantially enhancing detection generalization across unseen generators. Compared with CLIP-LoRA, DGS-Net improves detection accuracy by 17.7% and 29.2% respectively, highlighting its practicality for large-scale, real-world AI-generated image forensics.
>
> **4) Open source commitment.** We explicitly commit to releasing the complete source code and trained model weights on GitHub upon the official publication of this paper.
>
> **Q3 & W3: Concerns regarding Figure 1 visualization.**
>
> **1) DGS-Net demonstrates superior visual separation.** As illustrated in Figure 1, except for PhotoMaker, DGS-Net provides more distinct and pronounced gaps between real and fake regions than CLIP-LoRA across the other three datasets. This qualitative observation is further supported by the ablation results in Table 4, where DGS-Net (Row 4) significantly outperforms plain LoRA fine-tuning (Row 1). Together, these indicate that DGS-Net learns a more robust and discriminative decision boundary for generalized AI-image detection.
>
> **2) The dispersed distribution represents the preservation of pre-trained priors.** Pre-trained CLIP provides strong semantic geometry; in Figure 1 (Row 1), the clustered red regions reflect this intrinsic prior.  In contrast, standard LoRA fine-tuning is more prone to catastrophic forgetting, which can induce representation collapse and overfitting. Therefore, our core motivation is to learn more generalized artifact representations while preserving useful prior knowledge. The visualization supports this goal:  DGS-Net achieves superior separation (Inter-class separability, degree of separation between red and blue areas) while maintaining the original prior geometry (Intra-class prior preservation, the dispersed pattern in red area), which is exactly what ensures our model’s superior generalization performance.
>
> **3) Experimental verification.** To verify that DGS-Net extracts more generalized and transferable artifact representations, we conducted a linear probing experiment (frozen backbone + fresh FC head). The results are shown in Table I. The frozen CLIP-LoRA backbone drops to 52.4% on AIGIBench (significantly below the original CLIP's 71.5%), while DGS-Net remains substantially stronger (76.7%). This confirms that DGS-Net effectively preserves the intrinsic pre-trained geometry while extracting generalized forensic features, significantly outperforming standard fine-tuning.
>
> **Table I. Linear Probing Results. (Frozen Backbone)**
> |Method|AIGCDetectBench(Acc/AP)|AIGIBench(Acc/AP)|
> |:-:|:-:|:-:|
> |CLIP|72.7/91.9|71.5/75.7|
> |CLIP-LoRA|82.4/98.5|52.4/85.6|
> |**DGS-Net**|**93.3/98.8**|**76.7/85.4**|
>
> **L1: Discussion.**
>
> In the revised version, we will add dedicated Limitations section. Specifically, we will discuss: remaining failure cases on hard subsets; additional training overhead.

---

### Decision · Program_Chairs · 2026-04-30

**Decision:**

Accept (spotlight)

**Comment:**

Based on five reviews and the authors’ rebuttal, I recommend **Accept**. The paper presents a novel and technically sound approach for fine-tuning CLIP for AI-generated image detection by decomposing gradients into harmful and beneficial directions via orthogonal suppression and prior alignment. While one reviewer raised concerns about performance saturation and presentation clarity, the rebuttal effectively addressed these with additional experiments (linear probing, AUC, robustness tests, computational cost analysis, and cross-captioner validation), and the other four reviewers significantly increased their confidence or scores. The method demonstrates strong empirical gains across 50 generative models, and the commitment to open-source code ensures reproducibility. Overall, the contribution is solid, timely, and likely to be built upon by the community.